# Metal 3D printing technology for functional integration of catalytic system

Qinhong Wei[1,2,7], Hangjie Li[1,7], Guoguo Liu[1,7], Yingluo He[1], Yang Wang[1], Yen Ee Tan[1], Ding Wang[3], Xiaobo Peng ®[4,5 ✉], Guohui Yang ®[1,6 ✉] & Noritatsu Tsubaki ®[1 ✉]

Mechanical properties and geometries of printed products have been extensively studied in metal 3D printing. However, chemical properties and catalytic functions, introduced by metal 3D printing itself, are rarely mentioned. Here we show that metal 3D printing products themselves can simultaneously serve as chemical reactors and catalysts (denoted as self-catalytic reactor or SCR) for direct conversion of C1 molecules (including CO, $CO_2$ and $CH_4$) into high value-added chemicals. The Fe-SCR and Co-SCR successfully catalyze synthesis of liquid fuel from Fischer-Tropsch synthesis and $CO_2$ hydrogenation; the Ni-SCR efficiently produces syngas (CO/$H_2$) by $CO_2$ reforming of $CH_4$. Further, the Co-SCR geometrical studies indicate that metal 3D printing itself can establish multiple control functions to tune the catalytic product distribution. The present work provides a simple and low-cost manufacturing method to realize functional integration of catalyst and reactor, and will facilitate the developments of chemical synthesis and 3D printing technology.

[1] Department of Applied Chemistry, School of Engineering, University of Toyama, Gofuku 3190, Toyama 930-8555, Japan. [2] Department of Chemical Engineering, School of Petrochemical Technology and Energy Engineering, Zhejiang Ocean University, Zhoushan 316022, China. [3] School of Material Science & Engineering, University of Shanghai for Science and Technology, Shanghai 200093, China. [4] National Institute for Materials Science, 1-1 Namiki, Tsukuba, Ibaraki 305-0044, Japan. [5] Key Laboratory of the Ministry of Education for Advanced Catalysis Materials, Institute of Physical Chemistry, Zhejiang Normal University, Jinhua 321004, China. [6] State Key Laboratory of Coal Conversion, Institute of Coal Chemistry, Chinese Academy of Sciences, Taiyuan 030001, China. [7]These authors contributed equally: Qinhong Wei, Hangjie Li, Guoguo Liu. ✉email: PENG.Xiaobo@nims.go.jp; thomas@eng.u-toyama.ac.jp; tsubaki@eng.u-toyama.ac.jp

Catalysts and reactors are two essential elements of traditional catalytic systems. Catalysts can change reaction pathway, improve reaction efficiency, or selectively produce target chemicals[1–3]. Reactors possess an important function that provides suitable environments for various catalytic reactions. Although the two essential elements have been developed for so many years, their research focuses were distinctly different. The catalyst researches mainly focused on preparation methods, reaction mechanisms, structure characterizations, catalyst performances, and so on[1,2]. However, the reactor researches were devoted to updating reactor types and functions, enhancing heat and mass transfer, lowering pressure drop, etc.[3] Until now, the researches of catalysts and reactors are still two different directions. Few studies have succeeded in functional integration of catalyst and reactor to effectively control chemical reaction. Therefore, there is a great need to develop their functional integration and synergies for future catalytic systems to realize superior chemical synthesis.

Three-dimensional (3D) printing has been widely studied in biotechnology, prosthetics, architecture, pharmaceutical synthesis, etc. (Fig. 1a)[4–14]. Recently, several research groups have also made considerable progress in catalyst preparation and reactor design[7,8,14–30]. The 3D printing techniques, such as direct ink writing (DIW)[15–24,28], fused deposition modeling (FDM)[14,16,26–28], stereolithography (SLA)[29] and selective laser sintering (SLS)[30], were employed and developed to print the functional catalysts or reactors. The printed catalysts or reactors have exhibited many new and exciting trends for chemical synthesis and analysis. However, the manufacturing principles of the catalysts and reactors were independent of each other. The synergies between them were also overlooked. In addition, the catalysts and reactors were printed separately. It could result in complicated printing process and low printing speed. In view of these problems, it is necessary to explore simple and fast manufacturing strategies. Metal 3D printing reactor, simultaneously coupled with catalytic function, is a feasible approach to overcome these obstacles. Moreover, it can be applied to harsh reaction conditions, such as high temperature and/or high pressure, as in huge catalytic facilities of petrochemical or C1 chemical complex.

Traditionally, petroleum refining is the main way to produce liquid fuel. But, with rapid depletion of petroleum reserves, it is urgent to develop new synthetic routes for conversion of non-petroleum resources (such as natural gas/shale gas, $CO_2$, biomass) into liquid fuel[31,32]. Fischer–Tropsch (FT) synthesis[33,34], $CO_2$ hydrogenation[35,36], and $CO_2$ reforming of $CH_4$ (DRM)[37,38], as the alternative routes or key steps, have been studied for a long time. But huge operating costs in the conventional plants always prevented them from large-scale industrial applications[39,40]. To effectively reduce the operating costs, metal 3D printing is a very promising technology to revolutionize the devices. On this basis, our integrated design, that is the printed self-catalytic reactors (SCRs) in combination with various catalytic functions (Fig. 1a, b), can further minimize the costs and reactor sizes dramatically, and improve the energy efficiency.

Herein, we design and manufacture three kinds of SCRs (Fe-SCR, Co-SCR, and Ni-SCR) to realize direct conversion of C1 molecules (including CO, $CO_2$, and $CH_4$) into high value-added chemicals. The Fe-SCR and Co-SCR produce highly selective synthesis of liquid fuel in high-pressure FT synthesis and $CO_2$ hydrogenation. The Ni-SCR generates high conversions of $CO_2$ and $CH_4$ in high-temperature DRM. Moreover, geometrical studies of the Co-SCRs demonstrate that metal 3D printing itself can enhance synergies between catalyst and reactor, and establish multiple control functions to tune the catalytic product distribution. We anticipate that these printing designs will facilitate the further development of 3D printing technology, and spark a technological revolution of traditional chemical industries and machinery manufacturing industries.

## Results and discussion

**Catalytic performance of SCRs**. Computer-assisted design (CAD, Rhinoceros 5.0) was utilized to create the virtual models of SCRs. To boost up the inner surface area, we designed a series of internal channels and semispherical bulges in the SCRs (Supplementary Fig. 1). The Fe-SCR, Co-SCR, and Ni-SCR were fabricated via SLS technology in the printing processes. Schematic diagram of the SLS technology was displayed in Supplementary

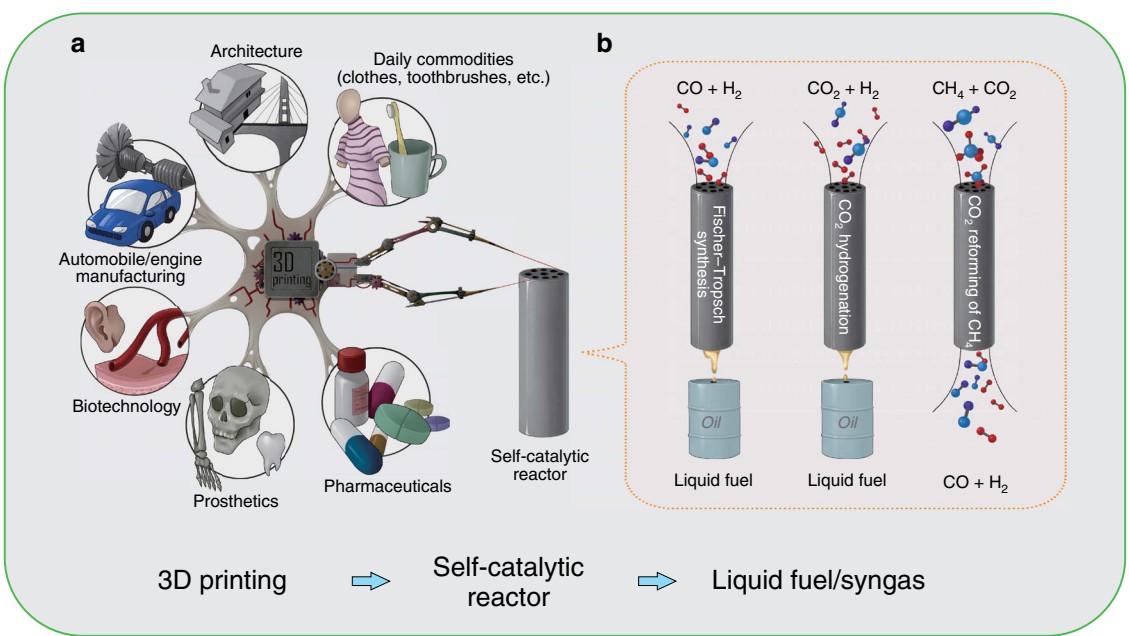

**Fig. 1 3D printing for self-catalytic reactor (SCR) and other typical applications. a** 3D printing for SCR and other typical applications. **b** The SCR for Fischer–Tropsch (FT) synthesis, $CO_2$ hydrogenation, and $CO_2$ reforming of $CH_4$ (DRM).

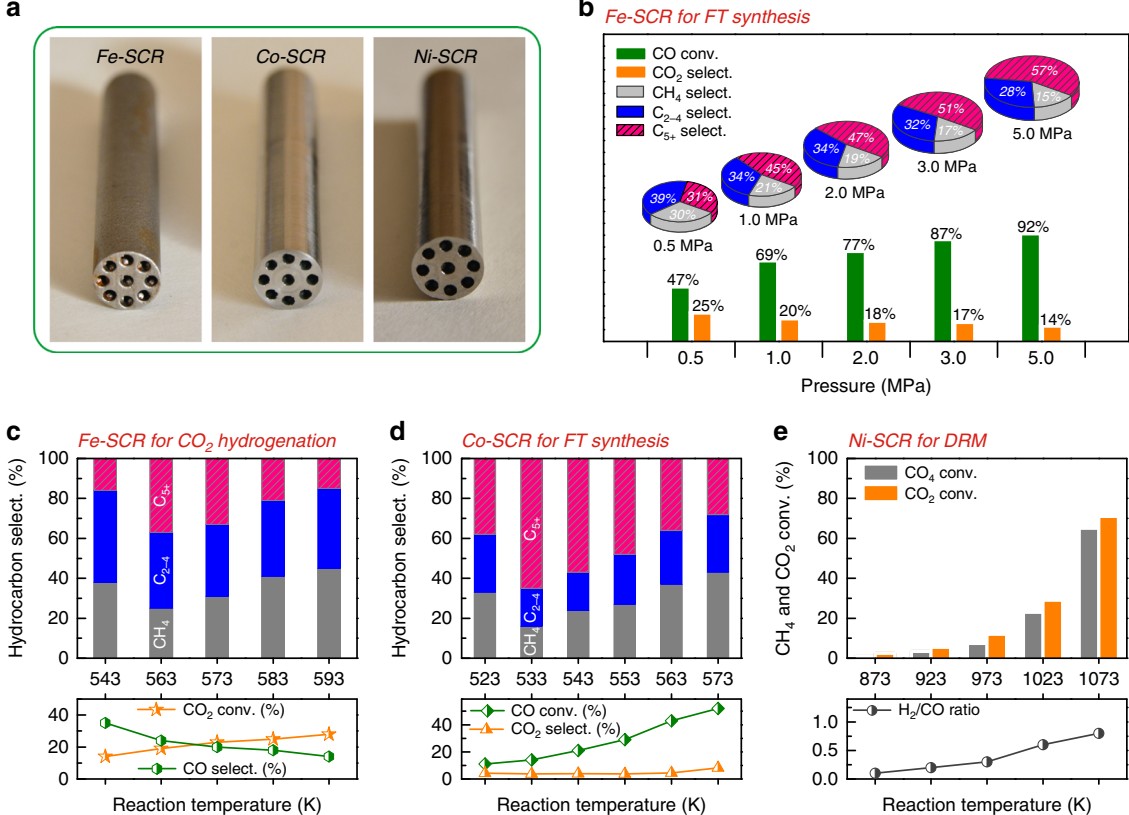

**Fig. 2 Catalytic performance of SCRs. a** The physical SCRs after polishing the outer surface. **b** Fe-SCR for Fischer–Tropsch synthesis. Reaction conditions: $T =$ 573 K; $H_2/CO = 2.0$; flow rate, 20 ml min$^{-1}$; time on stream, 10 h. **c** Fe-SCR for $CO_2$ hydrogenation. Reaction conditions: $P = 1.0$ MPa; $H_2/CO_2 = 3.0$; flow rate, 20 ml min$^{-1}$; time on stream, 10 h. **d** Co-SCR for Fischer–Tropsch synthesis. Reaction conditions: $P = 2.0$ MPa; $H_2/CO = 2.0$; flow rate, 20 ml min$^{-1}$; time on stream, 10 h. **e**, Ni-SCR for $CO_2$ reforming of $CH_4$. Reaction conditions: $P = 0.1$ MPa; $CO_2/CH_4/Ar = 45/45/10$; flow rate, 40 ml min$^{-1}$; time on stream, 8 h.

Fig. 2. The printing processes were further shown in Supplementary Movie 1 (see Methods for more printing details). Although the printing feedstocks of Fe-, Co- and Ni-based alloy powders (denoted as Fe-, Co-, and Ni-Powder) possessed remarkably different elemental compositions (see Supplementary Tables 1–3), the resulting SCRs still showed the same color and high fidelity (Fig. 2a).

To evaluate the catalytic ability, we first employed the Fe-SCR in FT synthesis. Before the reaction, the Fe-SCR was simply calcined in air and reduced by hydrogen. The obtained samples were denoted as Fe-SCR-*Calcined* and Fe-SCR-*Reduced*, respectively. Fe-Powder, as an allotropic catalyst, was also pretreated using the same calcination and reduction processes. We labeled the resulting samples with Fe-Powder-*Calcined* and Fe-Powder-*Reduced*, respectively. The nitrogen ($N_2$) physisorption results demonstrated that these pretreatment processes did not significantly change BET areas for the Fe-SCRs or Fe-Powders (Supplementary Table 4). The Fe-SCRs generated low BET areas, because bulk phase of the Fe-SCRs was highly dense and did not contribute to the $N_2$ physisorption. Traditional reaction tube ($T_{316}$) was purchased from the Swagelok Company. The $T_{316}$ and the Fe-Powder were also evaluated in FT synthesis. The schematic of FT reaction devices was displayed in Supplementary Fig. 3 and the reaction results were compared in Supplementary Table 5.

The $T_{316}$, produced by subtractive manufacturing, did not show any catalytic ability for the FT synthesis, probably because it was difficult to activate CO molecules on the smooth and dense surface (Supplementary Figs. 4a, b). The Fe-Powder displayed a high CO conversion of 74%, but the undesired $CH_4$ and $C_{2-4}$ were the main products. Note that the Fe-SCR also exhibited high CO

conversion. Moreover, the Fe-SCR possessed a higher liquid fuel selectivity of $C_{5+}$ than that of the Fe-Powder. In addition, we also compared linear velocity ($V_{linear}$) of syngas for the $T_{316}$, Fe-Powder and Fe-SCR (Supplementary Table 6). The linear velocity ($V_{linear}$) is defined as translational velocity of syngas, i.e., $V_{linear} = F_{CO+H2} / S_{cat}$. ($F_{CO+H2}$ and $S_{cat}$ represent flow rate of syngas and catalyst surface, respectively). The results unveiled that both the Fe-Powder and the Fe-SCR exhibited lower $V_{linear}$ than the traditional reaction tube $T_{316}$.

Different from subtractive manufacturing principle, metal 3D printing technology produces the SCRs via a layer-by-layer way, as shown in the Supplementary Movie 1. Thus, high-pressure tolerance of the SCRs is an important checkpoint. Based on this consideration, we dramatically raised the pressure from 0.5 to 5 MPa in FT synthesis (Fig. 2b). Nonetheless, our Fe-SCR system still worked well. The FT results exhibited that the increasing pressures enhanced the CO conversion and $C_{5+}$ selectivity, and inhibited the undesired formation of $CH_4$, $C_{2-4}$ and $CO_2$. To further confirm the reliability, we tuned the reaction temperatures and reused the Fe-SCR for 5 times in the FT synthesis (Supplementary Figs. 5, 6). The test results also suggest that the Fe-SCR possesses a wide range of operating temperature and high reusability.

We applied scanning electron microscope (SEM) to investigate the inner microstructure of the Fe-SCR. The results clearly displayed that the Fe-SCR had a grainy inner surface (Supplementary Fig. 4c, d), and significantly differed from the $T_{316}$. Since Fe-Powder was the feedstock and allotropic catalyst of the Fe-SCR, we further characterized the Fe-Powder before and after the pretreatments, to elucidate the origin of the catalytic ability. The

SEM observation uncovered that although the surface morphology of Fe-Powder was different from that of Fe-Powder-*Calcined*, the average particle sizes still kept at around 18 μm (Supplementary Fig. 7). More importantly, the X-ray diffraction (XRD) patterns and transmission electron microscopy (TEM) results, on the Fe-Powder and Fe-Powder-*Calcined*, showed that the phases of iron alloys were mainly transformed into α-$Fe_2O_3$ and $Fe_3O_4$ via the calcination process (Supplementary Figs. 8–12). Their [57]Fe Mössbauer spectra and Raman spectra analyses also demonstrated that the α-$Fe_2O_3$ and $Fe_3O_4$ were the main phases after the calcination (Supplementary Figs. 13–15 and Supplementary Table 7)[41–46]. According to these analyses of the allotropic catalyst, we consider that the grainy inner surface of the Fe-SCR with the abundant iron oxide species is critical to subsequent FT synthesis.

We also compared the chemical properties of Fe-SCR and Fe-Powder after the pretreatments, to reveal the influence of SLS process on the Fe-SCR. The XRD patterns of Fe-SCR and Fe-SCR-*Calcined*, showed that the iron alloy phases were also transformed into α-$Fe_2O_3$ and $Fe_3O_4$ (Supplementary Fig. 16). Although the intensities of XRD peaks were very week, they still displayed the same iron phases as the Fe-Powder and Fe-Powder-*Calcined*, respectively (Supplementary Fig. 8). The energy dispersive X-ray spectroscopy (EDS) analyses further unveiled that the element distributions of Fe-SCR and Fe-SCR-*Calcined* were similar to those of Fe-Powder and Fe-Powder-*Calcined*, respectively (Supplementary Table 8). Moreover, the X-ray photoelectron spectroscopy (XPS) analyses of Fe 2*p* region, demonstrated that the Fe-SCR and Fe-Powder after the pretreatments also possessed similar surface chemical states of the iron species (Supplementary Fig. 17a, b)[47–50]. Based on these analyses, we confirm that the SLS process of Fe-SCR did not obviously affect their surface chemical properties.

Through the calcination process and FT synthesis, the Fe-SCR changed the color from silver to black (Supplementary Fig. 18a). Because it was difficult to directly characterize the spent Fe-SCR (Fe-SCR-*Spent*) by XRD and [57]Fe Mössbauer spectra, we utilized the spent Fe-Powder (Fe-Powder-*Spent*) as a reference sample in the characterizations. The XRD and [57]Fe Mössbauer spectra results demonstrated that large part of the iron oxide was further transformed into χ-$Fe_5C_2$ over the Fe-Powder-*Spent* (Supplementary Figs. 19 and 20 and Supplementary Table 7), which is a well-recognized catalytic center for FT synthesis[50–52]. This observation implies that highly active catalytic center can be readily formed in the Fe-SCR.

To further investigate the highly active iron carbide, we cut the Fe-SCR and Fe-SCR-*Spent* into small slices (Supplementary Fig. 18b), and analyzed cross-section of the slices. The SEM analyses showed that the cross-section of Fe-SCR was monolithic, but a new layer on the inner surface of Fe-SCR-*Spent* was formed (Supplementary Fig. 21a, c). The EDS results demonstrated that the new layer possessed much higher carbon content than bulk phase of Fe-SCR-*Spent* and the cross-section of Fe-SCR (Supplementary Fig. 21b, d). It indicates that the new layer was carburized by FT process. We further employed temperature-programmed oxidation technique ($O_2$-TPO) to measure the carbon content on the Fe-SCR and Fe-SCR-*Spent*[53,54]. As in Supplementary Fig. 22, the peak I and peak II in the $O_2$-TPO profiles should be due to inherent carbon species and newly formed carbon species, respectively. According to the $O_2$-TPO profiles, we calculated the carbon content for the Fe-SCR-*Spent*. The results manifested that the total carbon content was 0.03 wt% of the Fe-SCR-*Spent*, and the carbon retention was 0.14 wt% of the carbon source of syngas (Supplementary Table 9).

We also utilized the XPS analysis to characterize the inner surface of the Fe-SCR-*Reduced* and Fe-SCR-*Spent*. The Fe 2*p*

peaks at 706.8 and 707.2 eV were identified on the Fe-SCR-*Reduced* and Fe-SCR-*Spent* (Supplementary Fig. 17b, c), and were attributed to iron metal and iron carbide[48–50], respectively. This proved that iron metal species were transformed into iron carbide species on the inner surface of Fe-SCR during the FT synthesis. The same phenomena were also observed on the Fe-Powder-*Reduced* and Fe-Powder-*Spent* in the XPS analysis (Supplementary Fig. 17b, c). Moreover, the XRD and [57]Fe Mössbauer spectra results of Fe-Powder-*spent* have demonstrated that the χ-$Fe_5C_2$ was the main phase during the reaction (Supplementary Figs. 19, 20 and Supplementary Table 7). Therefore, we confirm that the grainy inner surface with the highly active χ-$Fe_5C_2$ on the Fe-SCR promotes the FT performance.

Unexpectedly, the Fe-SCR also exhibited a good performance for $CO_2$ hydrogenation to liquid fuel, as shown in Fig. 2c. In the hydrogenation, high CO selectivity but low $CO_2$ conversion were formed in the low-temperature reaction. It indicates that reverse water-gas shift (RWGS: $CO_2 + H_2 = CO + H_2O$) reaction dominated the hydrogenation process[36,55,56]. In contrast, low CO selectivity but high $CO_2$ conversion were realized during the high-temperature reaction. This suggests that the generated CO from the RWGS was in situ converted into hydrocarbons via CO hydrogenation[36,55,56]. But high $CH_4$ selectivity was caused by the excessive high temperature. Consequently, by the reaction at 563 K, we obtained the moderate $CO_2$ conversion of 19% and high $C_{5+}$ selectivity of 37% over the Fe-SCR.

In addition to the Fe-SCR, we also fabricated the Co-SCR for the FT synthesis (see Supplementary Fig. 23 for inner surface morphology). The results were shown in Fig. 2d. By tuning the reaction temperature, the CO conversion significantly increased from 11 to 52%. The undesirable $CO_2$ selectivity was about 5%, and much lower than that of the Fe-SCR. Moreover, the $C_{5+}$ selectivity could reach 65 %. The inductively coupled plasma atomic emission spectroscopy (ICP-AES) analysis demonstrated no metal contamination in the liquid fuel (Supplementary Table 10). These findings indicate that the Co-SCR not only enhances the liquid fuel selectivity, but also inhibits the formation of $CO_2$ by-product. Additionally, we compared the thermal conductivity of the Co-SCR and traditional supports ($SiO_2$ and $Al_2O_3$) of FT catalysts. As indicated in Supplementary Table 11, the thermal conductivity of the Co-SCR was 7 W m$^{-1}$ K$^{-1}$, which was much higher than those of the traditional $SiO_2$ and $Al_2O_3$. It suggests that local overheating that leads to deactivation of the traditional FT catalysts, can be significantly reduced on the SCRs system[57,58]. More importantly, the high thermal conductivity can greatly improve energy efficiency for the whole reaction system.

To demonstrate high-temperature tolerance of our SCRs, we evaluated the Ni-SCR in DRM reaction (DRM: $CH_4 + CO_2 = 2CO + 2H_2$), at different reaction temperatures. Because FT synthesis and $CO_2$ hydrogenation are always performed at a relatively low reaction temperature, but DRM is a typical high-temperature reaction (~1000 K), due to the extremely inert feedstocks ($CO_2$ and $CH_4$)[59,60]. The DRM results were shown in Fig. 2e. At a high temperature of 1073 K, the $CH_4$ and $CO_2$ conversions as high as 65 and 71% were obtained, respectively. The ratio of $H_2$/CO was close to the stoichiometric ratio of 1. This observation indicates that the Ni-SCR suppresses the side reactions (RWGS and $CH_4$ decomposition ($CH_4 = C(s) + 2H_2$))[61,62], and keeps a good balance on the formation of CO and $H_2$. It also signifies that the Ni-SCR successfully resists the high-temperature reaction environment.

**Geometrical studies of Co-SCRs.** To reveal high degree of flexibility and freedom for our SCRs designs, we conducted the geometrical studies on the Co-SCR. The Co-SCR was further

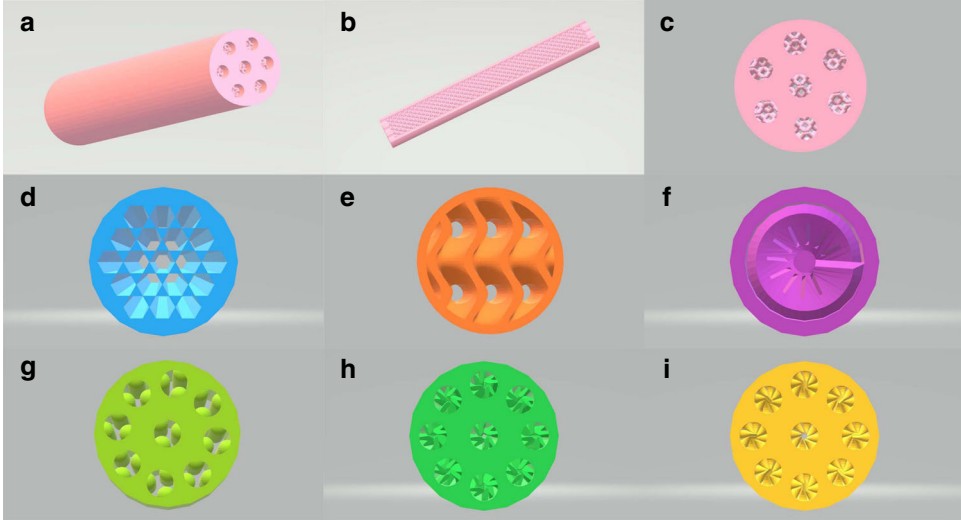

**Fig. 3 Geometrical structures of Co-SCRs.** (**a**) Co-SCR-1; (**b**) longitudinal section of Co-SCR-1; (**c**) cross-section of Co-SCR-1; (**d**) cross-section of Co-SCR-2; (**e**) cross-section of Co-SCR-3; (**f**) cross-section of Co-SCR-4; (**g**) cross-section of Co-SCR; (**h**) cross-section of Co-SCR-5; (**i**) cross-section of Co-SCR-6.

employed in the studies, due to the fact that it has already exhibited the high liquid fuel selectivity in the FT synthesis. In addition, via using the CAD, we designed the other 6 types of Co-SCRs to tune the inner geometrical structures (denoted as Co-SCR$x$, $x = 1, 2, 3…,6$; see Fig. 3 and Supplementary Figs. 24, 25). The various morphologies of Co-SCRs were selected in the metal 3D printing, based on the rules of easy realization, tunable geometrical structure, and high tolerance of pressure/temperature. After the printing fabrications, these Co-SCRs were also applied in FT synthesis, to investigate the catalytic performance. We anticipated that the different geometrical designs would optimize the product distribution in the FT synthesis.

The FT results over the seven types of Co-SCRs were summarized, and compared in Fig. 4. Although the Co-SCRs displayed the similar CO conversion, $C_{ole}/C_n$ ratio and $C_{iso}/C_n$ ratio (Fig. 4a), their product selectivities were significantly different. In particular, the $C_{5+}$ selectivity over these Co-SCRs increased from 48 to 73% (Fig. 4b). We also compared the fuel selectivities of gasoline ($C_{5-11}$), jet fuel ($C_{8-16}$) and diesel fuel ($C_{10-20}$), on the Co-SCR-1, Co-SCR-3, and Co-SCR-6 (Fig. 4c). The results exhibited that the Co-SCR-6 generated higher selectivities to the jet fuel and diesel fuel, than the Co-SCR-1 and Co-SCR-3. These findings indicate that these controllable 3D printing structures successfully tune FT product distribution, and increase liquid fuel selectivity. Furthermore, the increase of chain growth probability (α) from 0.57 to 0.86 also suggests that they can promote growth of heavy hydrocarbons, and suppress undesired formation of light hydrocarbons (Fig. 4d)[63,64]. Therefore, we conclude that these tailor-made 3D printing designs can dramatically improve the product distribution, and enhance the catalytic functions.

To obtain the underlying reasons of influencing the liquid fuel selectivity, we further analyzed the inner surface and channel volume, and calculated the liner velocity ($V_{linear}$) and passage time ($T_{passage}$) of syngas for the Co-SCRs (Supplementary Table 12). The passage time is defined as the time it takes for syngas to pass the channel, i.e., $T_{passage} = V_{channel} / F_{CO+H2}$ ($V_{channel}$ and $F_{CO+H2}$ are the channel volume and flow rate of syngas, respectively). Because the $F_{CO+H2}$ of feed gas was constant in our FT evaluation, the high inner surface and low channel volume of Co-SCRs led to low $V_{linear}$ and $T_{passage}$, respectively.

It is well known in FT synthesis that large catalytic surface area and low linear velocity will enhance re-adsorption of intermediate α-olefin, to promote new carbon-chain growth[65,66]. Low channel volume and low passage time can reduce secondary reactions (hydrocracking/hydrogenolysis) of long-chain hydrocarbons[33,67], and keep high liquid fuel selectivity in FT process. Moreover, spatial structure of FT reactor plays a key role on regulating the balance between plug-flow and back-mixing modes of reaction gas[68,69]. Therefore, although the Co-SCRs revealed non-linear changes on the inner surface and channel volume (Supplementary Table 12), they still displayed a linear increase on the liquid fuel selectivity (Fig. 4b). It demonstrated that the multiple factors, as mentioned above, worked simultaneously, resulting in the non-linear, overlying phenomena of the factors. These analyses also proved that the SCRs designs can provide three kinds of tunable factors, including inner surface, channel volume and spatial structure, to realize high controllability on chemical synthesis.

In summary, we have successfully designed three kinds of SCRs (Fe-SCR, Co-SCR, and Ni-SCR) via metal 3D printing technology. The C1 catalytic results demonstrated that they can effectively resist the harsh reaction conditions, such as high temperature and high pressure. The Fe-SCR and Co-SCR exhibited excellent performance for FT synthesis and $CO_2$ hydrogenation to synthesize liquid fuel. The Ni-SCR displayed high conversions and ideal ratio of $H_2/CO$ for $CO_2$ reforming of $CH_4$. Further, the geometrical studies of the Co-SCRs clearly revealed that different printing structures can dramatically tune their catalytic functions. This work offers a simple, fast and feasible technology to build functional integration and synergies between reactors and catalysts, and facilitate the new designs of future catalytic system. We expect that this self-catalytic design can stimulate new developments for 3D printing technologies, and wide applications in chemistry, energy, pharmacy, material synthesis, machinery manufacturing, etc.

## Methods
**Preparation of SCR**. The virtual SCRs were created by CAD (Rhinoceros 5.0). The physical SCRs were prepared by metal 3D printing via a SLS. To print Fe-SCR, the Fe alloy powder was first packed in a raw material vat, and the base plate was mounted under the laser source. Then, the Fe alloy powder was rapidly shaped into the physical Fe-SCR on the base plate via transformation of the virtual SCRs. After this process, the residual Fe powder was removed by a vacuum cleaner. The Fe-SCR was obtained by further dismounting the base plate and polishing the outer surface. The fabrication processes of Fe-SCR were conducted by J-3D Co., Ltd. in Japan.

The preparation processes were filmed in Supplementary Movie 1. Via using the same preparation method, we further fabricated Co-SCR and Ni-SCR. To tune the inner surface of the Co-SCR, we applied the CAD to design different structures.

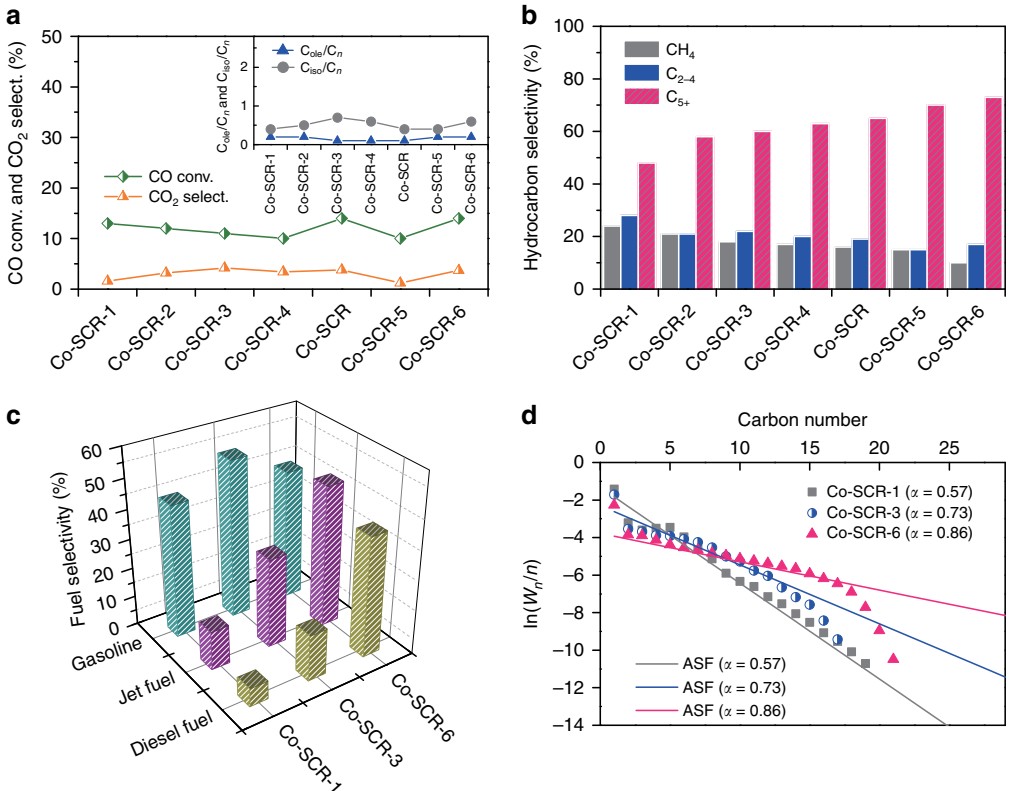

**Fig. 4 Impact of geometrical structures on FT product distribution. a** CO conversion, $C_{ole}/C_n$ ratio and $C_{iso}/C_n$ ratio of the Co-SCRs. **b** Hydrocarbon distribution of the Co-SCRs. **c** Fuel selectivity of gasoline ($C_{5-11}$), jet fuel ($C_{8-16}$), and diesel fuel ($C_{10-20}$) for the Co-SCR-1, Co-SCR-3, and Co-SCR-6. **d** Chain growth probability ($\alpha$) of the Co-SCRs. The $\alpha$ value was calculated based on the growth trend of hydrocarbons $C_{5-11}$ over the Co-SCR-1, Co-SCR-3, and Co-SCR-6, respectively. The calculation rule of the $\alpha$ value was according to the Anderson–Schulz–Flory model[63,64]. Reaction conditions: $T = 533$ K; $P = 2.0$ MPa; $H_2/CO = 2.0$; flow rate, 20 ml min$^{-1}$; time on stream, 10 h.

The other 6 types of Co-SCRs were also obtained using the same printing processes. The fabrication processes of Co-SCR and Ni-SCR were performed by Shanghai Consu Metal Materials Co., Ltd. in China.

To further activate the catalytic centers, the SCRs were simply pretreated before the reactions. The Fe-SCR was calcined in air at 873 K for 60 h, and then reduced by $H_2$ at 673 K for 10 h with a $H_2$ flow rate of 40 ml min$^{-1}$. The Co-SCRs were calcined in air at 1073 K for 24 h, and their inner surfaces were treated by an acid mixture of HCl (8.7 mol l$^{-1}$) and HNO$_3$ (3.3 mol l$^{-1}$) with a volume of 160 ml for 2 h. They were also reduced by the same $H_2$ conditions as those for the Fe-SCR. The Ni-SCR was calcined in air at 873 K for 60 h, and further reduced by $H_2$ at 1073 K for 6 h with the flow rate of 40 ml min$^{-1}$. Additionally, traditional reaction tube ($T_{316}$) was prepared from fully annealed and cold drawn stainless steel tube, which was purchased from the Swagelok Company (Type 316 L; Part No: SS-T6-S-049-20). The $T_{316}$ was pretreated by the same procedures as the Fe-SCR. In control experiments, the alloy powder was used as the reference samples and denoted as $M$ ($M =$ Fe, Co, Ni)-Powder. The Fe-Powder was also pretreated by using the same calcination and reduction processes as the Fe-SCR. Finally, these resulting Fe-SCR, Co-SCRs, Ni-SCR, $T_{316}$, and Fe-Powder (0.5 g) were used in C1 catalytic reactions.

**Structural characterization**. The XRD patterns were conducted on an X-ray diffractometer (RINT 2400; Rigaku) with a Cu-K$_\alpha$ radiation (40 kV and 40 mA). The elemental composition of the alloy powder was determined by a Philips Magix-601 wave-dispersive X-ray fluorescence (XRF) spectrometry. The surface morphology of the alloy powders was obtained by a scanning electron microscopy (SEM; JSM-6360LV; JEOL) with an energy-diffusive X-ray spectroscopy (EDX; JED-2300; JEOL). The transmission electron microscopy (TEM) images and high-resolution transmission electron microscopy images were obtained on a JEM-2100F (JEOL) at an acceleration voltage of 200 kV. The $^{57}$Fe Mössbauer spectra at room temperature were recorded on a Topologic 500 A spectrometer with a $^{57}$Co (Rh) source in a constant acceleration mode. The obtained spectra were fitted using MossWinn 4.0pre program, and the isomer shift values were given relative to $\alpha$-Fe as the standard. The Raman spectra were carried out on a Renishaw inVia 2000 Raman microscope by using an Ar ion laser at a wavelength of 514.5 nm. The nitrogen physisorption was carried out on a Micromeritics 3Flex surface characterization analyzer. In the analyses, SiO$_2$ pellet was used as an inner standard.

The Fe-SCR samples were cut into small pieces, and then loaded into the physical adsorption analyzer. The X-ray photoelectron spectroscopy (XPS) was performed on a Thermo Fisher Scientific ESCALAB 250Xi instrument with an Al K$\alpha$ X-ray radiation source. The temperature-programmed oxidation (O$_2$-TPO) was measured on a BELCAT II instrument equipped with an online mass spectrometer (BELMASS, BEL Japan, Inc., Japan). Before the test, the sample of 0.8 g was pre-treated with Ar gas (25 ml min$^{-1}$) at 393 K for 1 h, and then cooled down to 323 K. After a gas mixture of O$_2$/Ar (5 vol.% O$_2$) with a flow rate of 50 ml min$^{-1}$ was introduced into the instrument, the sample was heated to 1173 K with a rate of 10 K min$^{-1}$. For the quantitative measurement, the TPO profiles were calibrated by measuring a known amount of active carbon under the same conditions. The inductively coupled plasma atomic emission spectroscopy (ICP-AES) was tested using a Thermo Scientific iCAP 6300 instrument.

**Evaluation of catalytic performance**. FT synthesis was performed on the Fe-SCR and Co-SCRs. The FT apparatus was fabricated by ourselves. After the $H_2$ pretreatments, the SCRs were cooled down to 423 K. Syngas with a $H_2/CO$ ratio of 2.0 and flow rate of 20 ml min$^{-1}$ was introduced into the SCRs. Then, the catalytic systems were raised to target temperature and pressure, respectively, to start the reactions. The reusability experiment was conducted on the same reaction conditions. Before the experiment, pretreatments of the spent Fe-SCR were the same as those of the fresh Fe-SCR. After the FT synthesis, the gas products were analyzed by two online gas chromatographies (Shimadzu GC-8A with thermal conductivity detector (TCD); Shimadzu GC-14B with flame ionization detector (FID)). The liquid hydrocarbons in the effluent were captured by an ice trap, and analyzed by an off-line gas chromatography (Shimadzu GC-2014 with FID). The hydrocarbon selectivity was calculated based on the C-moles of a product with respect to the total C-moles in the hydrocarbon mixture. The calculation methods for conversion and selectivity are as follows[31,70]:

$$CO\ conv.\ \% = \frac{CO_{in} - CO_{out}}{CO_{in}} \times 100\%, \tag{1}$$

$$CO_2\ select.\ \% = \frac{CO_{2out}}{CO_{in} - CO_{out}} \times 100\%, \tag{2}$$

$$C_n \text{ select.} \% = \frac{nC_n}{\sum_{n=1}^{\max} nC_n} \times 100\%. \qquad (3)$$

$CO_2$ hydrogenation was also conducted on the Fe-SCR. The apparatus was the same with that of the FT synthesis. After the $H_2$ pretreatments, the Fe-SCR was cooled down to 423 K. Then, the feed gas with a $H_2/CO_2$ ratio of 3.0 and flow rate of 20 ml min$^{-1}$ was introduced, and the temperature and pressure were increased to the target conditions. The product analyses were similar with those of the FT synthesis. The calculation methods for conversion and selectivity are as follows[35,55]:

$$CO_2 \text{ conv.} \% = \frac{CO_{2in} - CO_{2out}}{CO_{2in}} \times 100\%, \qquad (4)$$

$$CO \text{ select.} \% = \frac{CO_{out}}{CO_{2in} - CO_{2out}} \times 100\%, \qquad (5)$$

$$C_n \text{ select.} \% = \frac{nC_n}{\sum_{n=1}^{\max} nC_n} \times 100\%. \qquad (6)$$

$CO_2$ reforming of $CH_4$ was carried out on the Ni-SCR at atmospheric pressure. A gas mixture of $CO_2$, $CH_4$ and Ar ($CO_2/CH_4/Ar = 45/45/10$ in vol.%) with a flow rate of 40 ml min$^{-1}$ was employed on this reaction. The reaction effluents were monitored using two online gas chromatographies. The Shimadzu GC-8A with TCD and Porapak Q column was used to analyze the CO, Ar, $CH_4$, and $CO_2$; The Shimadzu GC-2014 with TCD and activated carbon column was employed to analyze the $H_2$. The calculation methods for conversion and $H_2/CO$ ratio are presented as follows[59]:

$$CH_4 \text{ conv.} \% = \frac{CH_{4in} - CH_{4out}}{CH_{4in}} \times 100\%, \qquad (7)$$

$$CO_2 \text{ conv.} \% = \frac{CO_{2in} - CO_{2out}}{CO_{2in}} \times 100\%, \qquad (8)$$

$$H_2/CO \text{ ratio} = \frac{H_{2out}}{CO_{out}}. \qquad (9)$$

## Data availability

Source data are provided with this paper. The other data supporting the findings of this study are available from the corresponding authors on reasonable request.

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

## Acknowledgements
This work was supported by the Japan Science and Technology Agency (MIRAI-JPMJMI17E2). We acknowledge Ronggang Fan (University of Toyama, Japan) for his assistance to accomplish this work.

## Author contributions
Q.W., H.L., and G.L. performed most of the experiments and analyzed the experimental data. Y.H., Y.W., and D.W. conducted the Raman spectroscopy, scanning electron microscopy, and transmission electron microscopy, respectively. Y.E.T. performed the X-ray diffraction. X.P., G.Y., and N.T. designed the research, analyzed the data and wrote the manuscript. All authors discussed the results and commented on the manuscript at all stages.

## Competing interests
The authors declare no competing interests.
