## [Peer Review File · Nature Communications]

Reviewers' comments:

Reviewer #1 (Remarks to the Author):

The authors describe an interesting approach to fabricate self-catalytic reactors (SCR) made by selective laser sintering of catalytically active alloy powders. As demonstrated by experiments the 3D printing approach provides control over Fischer-Tropsch (FT) conversion rate and selectivity by changing the design of the 3D printed SCR. While this concept is very interesting and will certainly be of interest to others in the community and the wider field, the manuscript has two main shortcomings: 1) although the authors printed and tested different reactor designs to demonstrate that the FT product distribution changes with the reactor design, there is no discussion of the particular reactor designs and why the product selectivity changed (other than a statement that the design may change the contact time); and 2) many of the claims made by the authors are simply not supported by the material and discussions provided in the manuscript and the supplementary information. The specific comments below regarding additional information, characterization and discussion of the results need to be addressed to support the authors' claims and to allow other researchers to reproduce the results. After this major revision, the manuscript may be considered for publication.

Specific comments:

- 1) To highlight their approach, the authors state that the T316 reaction tube was produced by subtractive manufacturing. While I could not check this statement (no product number was provided) tubes are typically manufactured by extrusion or drawing.
- 2) What is the porosity and the specific surface area of the laser sintered material, before and after the calcination pretreatment? This information is critical for comparing catalytic properties of the SCR with the precursor alloy powders.
- 3) What is the surface area and volume of the reactor as defined by the channel design, as printed as well as after the calcination/hydrogen treatment?
- 4) What is the particle size and the specific surface area of the metal powders used for catalytic experiments (after the pretreatment), and how much powder was used in these tests? The SEM image shown in Figure S9b and the TEM image shown in S11a/b suggest a dramatic morphology change during the pretreatment.
- 5) What is the elemental surface composition of the SCRs and the metal powders after the described pretreatment method? It is not clear (but assumed by the authors) that the same pretreatment method for SCRs and metal powders leads to the same surface composition as new alloy phases may be formed during the laser sintering process which respond differently to the pretreatment. This requires the use of surface sensitive characterization techniques like photoelectron spectroscopy as it is the surface composition and not the bulk phase that controls the catalytic performance.

6) What is the measured conversion of the reactants CO (FT), CO₂ (CO₂ hydrogenation) and CO₂/CH₄ (CO₂ reforming of CH₄) into hydrocarbons? The authors report only how much of the reactants is missing in the outlet, but not to which amount the missing reactants were converted into hydrocarbons. The Moessbauer results summarized in Figure S17 and Table S5 and the TEM results in Figure S11d reveal the formation of Fe carbide phases, so certainly some of the missing carbon was deposited as carbide and not converted into hydrocarbons. The question is how much of the carbon from the reactants remains in the reactor in the form of carbides or carbon, and how much is transformed into the desired hydrocarbon products?

7) Figure S18 needs larger scale bars

8) The authors need to provide more information regarding the reusability experiment shown in Figure S6. Specifically, was any pretreatment performed before reusing the Fe-SCR?

9) What is the channel volume of the different SCR designs, and why were these designs chosen? The authors only stated that they designed different SCRs to tune the inner surface (page 5). Next the authors speculate that the design will affect the contact time of FT reactants and products. The catalyst/reactor contact time will certainly change with the channel volume if the flow rate is not accordingly adjusted. The contact time can be easily calculated given the known flow rate and the channel volume from the CAD design. How does the SCR contact time compare to the contact time in the T316/metal powder experiments? May be this discussion can be used to generate some general design rules?

Reviewer #2 (Remarks to the Author):

The paper is describing use of Selective Laser Sintering (or Selective Laser Melting, SLM) method for preparation catalysts for petroleum refining. The basic idea here is combination of the functionality of the printed object, in this case the reaction vessel, with the functionality of the printing material, which in this case serve as catalytically active material. This is not a new idea but one of the emerging trends in 3D printing. It is possible to have considerable synergetic advances when the object has dual functionality. It is possible to optimize the shape and size of the printed object and when it is printed with active material it is possible to select or optimize also the functionality of the material. Both of these are nicely presented in the paper.

I think this paper deserves to be published in Nature Communication. However, I think that the authors should note it the text that use of SLM-technique for printing catalytically active objects is part of a new and exciting trend in 3D printing, which aims at dual functionality of the printed object. Printing catalysts itself is not new. There are examples where FDM or similar type extruding techniques has been used to build catalytically active flow through objects: DOI: 10.1016/j.jcat.2015.11.019, DOI: 10.1016/j.cattod.2019.06.026, DOI: 10.1016/j.jclepro.2018.12.274, DOI: 10.1016/j.apcatb.2019.01.008,

DOI: 10.1149/2.0341905jes, DOI: 10.1016/j.jcou.2019.07.013, DOI: 10.1021/acsapm.9b00598.

In addition to these examples also SLS printing has been used for building reactor parts. Similarly, idea of printing catalytic reactors or reactor parts are well known: DOI: 10.1039/C7CY00615B, DOI: 10.1021/acscatal.7b02592, DOI: 10.1021/acsomega.9b00711. The last one is an example of use of SLS-technique for printing mixer for a catalyst reactor, but even in that one it has been mentioned that it would be possible to use printing for preparing any shape of objects. This means that the catalytic activity could be added in any part of the reaction vessel, not just in the walls of the reactor as in the paper here. This approach goes far beyond the catalysis. Similar technique could be used to prepare all kind of multifunctional objects. In addition to the biological systems it has been applied in various different areas. For example, NLO active lenses, where optical properties of the lens have been combined with NLO activity of the printing material, have been printed by using SLA technique (10.1021/acsomega.8b01659).

As a summary, I think the paper should be published in Nature communication. To my knowledge it is the first example of use of SLM printing to build catalytically active reactor vessels. It extends the use of 3D printed chemically functional objects into area of high pressure high temperature tasks. However, it should also be kept in mind that despite the obvious benefits of using metal printing it also has its limitations. It is not suitable for direct printing molecular materials i.e. molecular catalysts and it is not likely to be the first choice if the catalysis require noble metals. Therefore, the authors should put this into the wider context of latest developments in 3D-printing: multifunctional objects.

Reviewer #3 (Remarks to the Author):

In my opinion, this is a very interesting article since it addresses various scientific and technological aspects currently relevant. On the one hand, 3D-printing, an emerging technology. Particularly innovative is the technique based on selective laser sintering (SLS) using metallic powder, for the elaboration of new monolithic-type reactors. On the other hand, the catalytic activity presented, very efficient and relevant. Therefore, 3D-printing fabrication of metal-based catalytic devices is a field of much interest in catalysis today.

The degree of novelty and interest in this article is high due to several reasons:

The own technique used for 3D-printing (selective laser sintering, SLS), the composition of the reactors (Fe, Ni, Co) and the type of application for which you want to apply. On the other hand, although 3D-printing technology is mostly applied to thermoplastic polymers or ceramic materials, little development still exists for metal 3D-printing, due to the high temperatures necessary in the process due to the high melting point of the metal.

Although the references provided are quite illustrative and significant within the field, I think that some more reference could be added to the text, regarding the preparation and application of monoliths containing metallic species on the surface through 3D-printing, particularly those that incorporate metal

on their surface. An example is our work related to the use of monoliths in multi-catalysis (Antonio S. Díaz-Marta, ACS-Catal., 2018, 8, 392-404). The reactor-catalysts presented in this work are in fact 3D monolitos.

Although catalysts of this type (metal monolithics) have already been described for different catalytic transformations, in this work it is applied for industrial purposes in three different types of high value transformations:

FT was performed on Fe-SCR and Co-SCR.

Co₂ hydrogenation on Fe-SCR.

Co₂ reforming of methane on Ni.

The catalysts show a high level of reusability.

That is why I consider this article as very interesting.

The work seems to be very well executed. Two different centers, one in Japan and the other in China, have coordinated the manufacturing of these devices. One catalyst was prepared in Japan (Fe-SCR) and two in China (Ni-SCR and Co-SCR).

Indeed, the design integrates the concept of catalyst and reactor in the same device. The characterization of the material is very complete. However, there are some aspects of this work that have generated in me some questions or doubts:

-Since the specific surface is a key aspect in these reactors (basically it is for any type of reactor, ceramic, metal-ceramic or ultraporous) I wonder why the authors have not provided any data about BET area (specific surface). This would help the rest of the researchers in the field to have a clear idea regarding the catalytic surface of these manufactured materials.

SEM images do indeed reveal a grainy surface, which is interesting from the point of view of a larger specific surface. Other data that accompany the characterization of the material such as XRD, RAMAN, Mössbauer, are very complete.

In any case, I would like to know if they have considered measuring BET area or if they have an approximate idea of the value m^2 / g of catalyst.

-Related to the grainy nature of the surface, it would be positive to know if the researchers carried out some type of experiment to evaluate the possible leaching of metallic particles to the reaction medium, which could eventually contaminate the liquid-fuel (ICP or similar experiment). This is not an essential question anyway, although it does give an idea of the robustness of the monolith-reactor.

- The variability in the catalytic activity as a consequence of the design in different geometries of the reactors is discussed in the article. The selectivity of Co-SCR-6 to generate gasoline, jet fuel or diesel fuel is particularly interesting. However, from the reading at the end of the article, it is not clear to me if this variation is as a consequence of an increase in the specific surface area of the reactor or rather the geometry of the reactor itself, a longer passage time of the reagents depending on the shape, or a combination of the two factors.

-As for the virtual design (CAD) of the catalyst-reactor, the software they have used is not specified.

- The supplied video is very illustrative of the SLS process.

I highly recommend publishing this article.

Responses to Reviewer Comments

Reviewer #1 (Remarks to the Author):

The authors describe an interesting approach to fabricate self-catalytic reactors (SCR) made by selective laser sintering of catalytically active alloy powders. As demonstrated by experiments the 3D printing approach provides control over Fischer-Tropsch (FT) conversion rate and selectivity by changing the design of the 3D printed SCR. While this concept is very interesting and will certainly be of interest to others in the community and the wider field, the manuscript has two main shortcomings: 1) although the authors printed and tested different reactor designs to demonstrate that the FT product distribution changes with the reactor design, there is no discussion of the particular reactor designs and why the product selectivity changed (other than a statement that the design may change the contact time); and 2) many of the claims made by the authors are simply not supported by the material and discussions provided in the manuscript and the supplementary information. The specific comments below regarding additional information, characterization and discussion of the results need to be addressed to support the authors' claims and to allow other researchers to reproduce the results. After this major revision, the manuscript may be considered for publication.

Response: We greatly thank Reviewer 1 for the guidance, and do appreciate the positive and constructive comments on our manuscript. They help us to gain insights into the SCR system, and also give us the greatest confidence to further continue this research. 3D printing technologies have been widely studied in many fields, such as biotechnology, prosthetics, etc. In the future, 3D printing technologies will also be a good trend in the fields of chemistry and chemical engineering. Metal 3D printing, as an important type of the technologies, possesses many inherent advantages. The printed metal devices can tolerate high temperatures and high pressures. The metal compositions are natural catalysts to various chemical reactions. Moreover, the precise fabrication via computer-assisted design (CAD) can guarantee high reusability, and

eliminate personal errors caused by different workers during the fabrication.

In the manuscript, we utilize the metal 3D printing to combine reactor function with catalyst function for conversion of CO, CO₂ and CH₄ into high value-added chemicals. It provides the first example for functional integration via metal 3D printing technology. We believe that in the future it will stimulate the developments of various 3D printing technologies to manufacture multi-functional devices, and have wide applications in fields of chemistry and chemical engineering.

Specific comments:

1. To highlight their approach, the authors state that the T316 reaction tube was produced by subtractive manufacturing. While I could not check this statement (no product number was provided) tubes are typically manufactured by extrusion or drawing.

Response: We appreciate the reviewer for this comment. We followed this suggestion, and have added the statements in the section of "Methods". The statements were also shown below:

“Additionally, traditional reaction tube (T₃₁₆) was prepared from fully annealed and cold drawn stainless steel tube, which was purchased from the Swagelok Company (Type 316L; Part No: SS-T6-S-049-20). The T₃₁₆ was pretreated by the same procedures as the Fe-SCR.”

2. What is the porosity and the specific surface area of the laser sintered material, before and after the calcination pretreatment? This information is critical for comparing catalytic properties of the SCR with the precursor alloy powders.

Response: We fully agree that the information of textural property is critical for comparing catalytic properties of the SCR with the powder-based catalyst. We followed this suggestion, and conducted N₂ physisorption on the Fe-Powder and Fe-SCR samples after the calcination, reduction and FT synthesis. The BET results were shown in Supplementary Table 4.

The BET results revealed that the Fe-Powder samples, after the calcination, reduction and FT synthesis, displayed almost the same surface area (12~13 m² g⁻¹). The

Fe-SCR samples, after the pretreatments and FT synthesis, also exhibited similar surface area (1.6~3.0 m² g⁻¹). The BET areas of Fe-SCR samples were lower than those of the Fe-Powder samples, because bulk phase of the Fe-SCRs was highly dense and did not contribute to the N₂ physisorption. In addition, we also added the statements in the "Results and discussion" of the manuscript, as follows:

“.....We labeled the resulting samples with Fe-Powder-*Calcined* and Fe-Powder-*Reduced*, respectively. The nitrogen (N₂) physisorption results demonstrated that these pretreatment processes did not significantly change BET areas for the Fe-SCRs or Fe-Powders (Supplementary Table 4). The Fe-SCRs generated low BET areas, because bulk phase of the Fe-SCRs was highly dense and did not contribute to the N₂ physisorption.....”

Supplementary Table 4 | BET area for Fe-Powder and Fe-SCR before and after the pretreatments.^a

Sample	BET area (m ² g ⁻¹) ^b
Fe-Powder	12
Fe-Powder- Calcined	12
Fe-Powder- Reduced	13
Fe-Powder- Spent	13
Fe-SCR	1.6
Fe-SCR- Calcined	2.6
Fe-SCR- Reduced	2.7
Fe-SCR- Spent	3.0

(a) SiO₂ pellet was used as an inner standard for the BET analysis. (b) The SCR samples were cut into small pieces, and the weight of each piece was about 0.20~0.25 g for the BET analysis.

3. What is the surface area and volume of the reactor as defined by the channel design, as printed as well as after the calcination/hydrogen treatment?

Response: We do appreciate the reviewer for these comments. As shown in Supplementary Table 4, we have measured the BET area of Fe-SCR samples after the calcination, reduction and FT synthesis. The results uncovered that the pretreatments and FT reaction did not significantly change the BET area of Fe-SCR samples, as

answered above. We also followed this suggestion and calculated the surface area and volume of the Fe-SCR via the CAD. The inner surface and channel volume of Fe-SCR were indicated in Supplementary Table 6.

Supplementary Table 6 | Linear velocity of syngas in the T₃₁₆, Fe-Powder and Fe-SCR.^a

Sample	Inner surface (10 ⁻³ m ²)	Channel volume (10 ⁻⁶ m ³)	Linear velocity ^c (10 ⁻⁴ m s ⁻¹)
T ₃₁₆	1.2	2.2	2.8
Fe-Powder	22 ^b	/	0.2
Fe-SCR	3.9	0.6	0.8

(a) Inner surface and channel volume were obtained according to the reactor size and CAD calculation. (b) The surface of Fe-Powder was calculated based on the average particle size. (c) The linear velocity was calculated based on the equation of $V_{linear} = F_{CO+H_2} / Scat.$ (V_{linear} , F_{CO+H_2} and $Scat.$ represent linear velocity of syngas, flow rate of syngas and catalyst surface, respectively). Syngas conditions: temperature, 573 K; pressure, 1.0 MPa; flow rate, 20 ml min⁻¹.

4. What is the particle size and the specific surface area of the metal powders used for catalytic experiments (after the pretreatment), and how much powder was used in these tests? The SEM image shown in Figure S9b and the TEM image shown in S11a/b suggest a dramatic morphology change during the pretreatment.

Response: We followed this suggestion. According to the BET results in Supplementary Table 4, the Fe-Powder samples, after the calcination, reduction and FT synthesis, displayed the surface area of 12~13 m² g⁻¹. The particle size distributions for the Fe-Powder and Fe-Powder-*Calcined* were analyzed and displayed in Supplementary Figure 7b, d. The average particle sizes of Fe-Powder and Fe-Powder-*Calcined* were about 18 μm.

In addition, the Fe-Powder of 0.5 g was used in the FT synthesis, and this statement was added in the section of "Methods".

“.....Finally, these resulting Fe-SCR, Co-SCRs, Ni-SCR, T₃₁₆ and Fe-Powder (0.5 g) were used in C1 catalytic reactions.”

The reviewer is right in that the SEM/TEM images of Fe-Powder (Supplementary Figure 10) and the TEM images of Fe-Powder-*Calcined* and Fe-Powder-*Spent* (Supplementary Figure 12) suggest a morphology change during the pretreatments. To clearly reveal the influence of the pretreatments, we further employed SEM analysis to compare the morphologies of Fe-Powder and Fe-Powder-*Calcined*. The SEM results, as shown in Supplementary Figure 7a, c, demonstrated that the Fe-Powder and Fe-Powder-*Calcined* formed different surface morphologies. However, their particle size distribution displayed that the average particle size was similar to each other. Based on these analyses, we have added the statement in the "Results and discussion" of the revised manuscript, as follows:

“.....we further characterized the Fe-Powder before and after the pretreatments, to elucidate the origin of the catalytic ability. The SEM observation uncovered that although the surface morphology of Fe-Powder was different from that of Fe-Powder-*Calcined*, the average particle sizes still kept at around 18 μm (Supplementary Fig. 7).....”

Supplementary Figure 7 | Surface morphology and particle size distribution for the Fe-Powder and Fe-Powder-*Calcined*. (a) SEM image for the Fe-Powder; (b) particle size distribution for the Fe-Powder; (c) SEM image for the Fe-Powder-*Calcined*; (d) particle size distribution for the Fe-Powder-*Calcined*. The average particle sizes of Fe-Powder and Fe-Powder-*Calcined* were about 18 μm .

5. What is the elemental surface composition of the SCR and the metal powders after the described pretreatment method? It is not clear (but assumed by the authors) that the same pretreatment method for SCR and metal powders leads to the same surface composition as new alloy phases may be formed during the laser sintering process which respond differently to the pretreatment. This requires the use of surface sensitive characterization techniques like photoelectron spectroscopy as it is the surface composition and not the bulk phase that controls the catalytic performance.

Response: We appreciate the reviewer for these comments. We followed the suggestions, and utilized EDS analysis to measure the surface elemental composition of the Fe-SCR and Fe-Powder before and after the calcination. The EDS results were shown in Supplementary Table 8. The element distribution of Fe-Powder and Fe-SCR were almost the same. The Fe-Powder-*Calcined* and Fe-SCR-*Calcined* also exhibited similar element distribution.

Supplementary Table 8 | EDS analysis for the Fe-Powder and Fe-SCR before and after the calcination.

Sample	Metal and non-metal contents (wt%)					
	C	O	Fe	Ni	Co	Others
Fe-Powder	0.7	0.8	66.7	17.3	8.7	5.8
Fe-SCR	3.4	1.1	65.7	16.0	8.9	4.9
Fe-Powder- Calcined	0.9	28.9	56.8	5.0	4.3	4.1
Fe-SCR- Calcined	3.2	27.2	53.0	8.6	4.1	3.9

The EDS analysis showed that the Fe-Powder and Fe-SCR possessed similar element distribution. The element distribution of Fe-Powder-*Calcined* was almost the same as that of the Fe-SCR-*Calcined*.

Supplementary Figure 16 | XRD patterns for the Fe-SCR and Fe-SCR-Calcined.

Supplementary Figure 17 | XPS spectra in Fe 2p region for the Fe-Powder and Fe-SCR after the pretreatments and FT reaction. (a) Fe 2p region for the Fe-Powder-*Calcined* and Fe-SCR-*Calcined*. The Fe 2p peaks at 711.2 and 724.6 eV were due to 2p3/2 and 2p1/2 for iron oxide, respectively. In general, Fe₃O₄ does not show satellite in Fe 2p region. The satellite at 719.4 eV should be attributed to the α -Fe₂O₃. The Fe 2p region with the weak satellites indicates a coexistence of α -Fe₂O₃ and Fe₃O₄. (b) Fe 2p region for the Fe-Powder-*Reduced* and Fe-SCR-*Reduced*. The peak at 706.8 eV was assigned to 2p3/2 of iron metal. The peak at 719.9 eV should be due to two overlapping components: the satellite 2p3/2 of α -Fe₂O₃ and 2p1/2 of iron metal. (c) Fe 2p region for the Fe-Powder-*Spent* and Fe-SCR-*Spent*. The peak at 707.2 eV corresponded to 2p3/2 of iron carbide. The peak at 720.2 eV was due to two overlapping components: the satellite 2p3/2 of α -Fe₂O₃ and 2p1/2 of iron carbide. The XPS analyses demonstrated that the Fe-Powder and Fe-SCR, after the pretreatments, showed similar Fe 2p regions.

We also utilized XRD analysis to directly characterize the Fe-SCR before and after the calcination. As shown in Supplementary Figure 16, the XRD results indicate that iron alloy phases of the Fe-SCR were also transformed into α -Fe₂O₃ and Fe₃O₄ on the Fe-SCR-*Calcined*. This phenomenon was the same as the Fe-Powder and Fe-Powder-*Calcined* (Supplementary Figure 8). We further employed XPS analysis to observe surface chemical states of the iron species on the Fe-Powder and Fe-SCR after the pretreatments. The results were displayed in Supplementary Fig. 17a, b. The XPS analysis on Fe 2p region demonstrated that the Fe-SCR and Fe-Powder, after the pretreatments, also possessed similar surface chemical states of the iron species. According to the XRD, EDS and XPS analyses, we confirm that the SLS process did not result in the formation of new alloy phases in the Fe-SCR. In the revised manuscript, we added the statement in the "Results and discussion", as shown below:

“We also compared the chemical properties of Fe-SCR and Fe-Powder after the pretreatments, to reveal the influence of SLS process on the Fe-SCR. The XRD patterns of Fe-SCR and Fe-SCR-*Calcined*, showed that the iron alloy phases were also transformed into α -Fe₂O₃ and Fe₃O₄ (Supplementary Fig. 16). Although the intensities of XRD peaks were very week, they still displayed the same iron phases as the Fe-Powder and Fe-Powder-*Calcined*, respectively (Supplementary Fig. 8). The energy dispersive X-ray spectroscopy (EDS) analyses further unveiled that the element distributions of Fe-SCR and Fe-SCR-*Calcined* were similar to those of Fe-Powder and Fe-Powder-*Calcined*, respectively (Supplementary Table 8). Moreover, the X-ray photoelectron spectroscopy (XPS) analyses of Fe 2p region, demonstrated that the Fe-SCR and Fe-Powder after the pretreatments also possessed similar surface chemical states of the iron species (Supplementary Fig. 17a, b)⁴⁷⁻⁵⁰. Based on these analyses, we confirm that the SLS process of Fe-SCR did not obviously affect their surface chemical properties.”

6. What is the measured conversion of the reactants CO (FT), CO₂ (CO₂ hydrogenation) and CO₂/CH₄ (CO₂ reforming of CH₄) into hydrocarbons? The authors report only how much of the reactants is missing in the outlet, but not to which amount the missing reactants were converted into hydrocarbons. The Mössbauer results summarized in Figure S17 and Table S5 and the TEM results in Figure S11d reveal the formation of Fe carbide phases, so certainly some of the missing carbon was deposited as carbide and not converted into hydrocarbons.

The question is how much of the carbon from the reactants remains in the reactor in the form of carbides or carbon, and how much is transformed into the desired hydrocarbon products?

Response: We greatly thank Reviewer 1 for this comment. The reviewer is right, because the formation of carbide species resulted in the consumption of carbon source of syngas. To measure the carbon amount of carbide species, we performed O₂-TPO analysis on the Fe-SCR and Fe-SCR-*Spent* samples. The O₂-TPO results were displayed in Supplementary Figure 22 and Table 9. In the O₂-TPO profiles, the peak I was observed on the Fe-SCR, indicating inherent carbon species. In addition to the peak I, peak II was also observed on the Fe-SCR-*Spent*, suggesting that new carbon species were formed in the FT synthesis. The total carbon content of the Fe-SCR and Fe-SCR-*Spent* was 0.01 wt% and 0.03 wt%, respectively. The carbon retention was 0.14 wt% of the carbon source of syngas. It implies that the carbon source of 99.8 wt% from the syngas reactant can be utilized in catalytic reaction.

Supplementary Figure 22 | O₂-TPO profiles for the Fe-SCR and Fe-SCR-*Spent*. In the O₂-TPO profiles, the peak I was observed on the Fe-SCR, indicating that inherent carbon species in the Fe-SCR were oxidized. In addition to the peak I, peak II was also observed on the Fe-SCR-*Spent*, suggesting that new carbon species were formed in the FT synthesis.

Supplementary Table 9 | O₂-TPO analysis for carbon content on the Fe-SCR and Fe-SCR-*Spent*.

Sample	Carbon from CO ₂ peak I (μg)	Carbon from CO ₂ peak II (μg)	Carbon from CO peak I (μg)	Carbon from CO peak II (μg)	Total carbon content in Fe-SCR (wt%)	Carbon retention from syngas (wt%)
Fe-SCR	77	/	5	/	0.01	/
Fe-SCR- Spent	110	141	8	3	0.03	0.14

The amount of carbon was calculated based on the O₂-TPO profiles. The analyses exhibited that the total carbon content on Fe-SCR and Fe-SCR-*Spent* was 0.01 wt% and 0.03 wt%, respectively. The carbon retention was 0.14 wt% of the carbon source of syngas.

Additionally, we further used the SEM and EDS to analyze the cross-section of Fe-SCR and Fe-SCR-*Spent*, to reveal the carbide species on the inner surface. The analyses were exhibited in Supplementary Figure 21a-d. We found that a new layer was formed on the inner surface of Fe-SCR-*Spent*. Moreover, the new layer had much higher carbon content than bulk phase of Fe-SCR-*Spent* and the cross-section of Fe-SCR. More importantly, the XPS analysis in Supplementary Fig. 17c demonstrated that the iron carbide was formed on the inner surface. The XRD and ⁵⁷Fe Mössbauer spectra results of Fe-Powder-*spent* (Supplementary Figures 19, 20 and Supplementary Table 7) also demonstrated that the χ -Fe₅C₂ was the main phase of the Fe-Powder-*spent*. Thus, we confirm that the grainy inner surface with the highly active χ -Fe₅C₂ on the Fe-SCR promotes the FT performance. In the revised manuscript, the detailed statements were added in the section of "Results and discussion", as follows:

“To further investigate the highly active iron carbide, we cut the Fe-SCR and Fe-SCR-*Spent* into small slices (Supplementary Fig. 18b), and analyzed cross-section of the slices. The SEM analyses showed that the cross-section of Fe-SCR was monolithic, but a new layer on the inner surface of Fe-SCR-*Spent* was formed (Supplementary Fig. 21a, c). The EDS results demonstrated that the new layer possessed much higher carbon content than bulk phase of Fe-SCR-*Spent* and the cross-section of Fe-SCR (Supplementary Fig. 21b, d). It indicates that the new layer was carburized by FT process. We further employed temperature-programmed oxidation technique (O₂-TPO) to measure the carbon content on the Fe-SCR and Fe-SCR-*Spent*^{53,54}. As in Supplementary Fig. 22, the peak I and peak II in the O₂-TPO profiles should be due to inherent carbon species and newly formed carbon species, respectively. According to the O₂-TPO profiles, we calculated the carbon content for the

Fe-SCR-*Spent*. The results manifested that the total carbon content was 0.03 wt% of the Fe-SCR-*Spent*, and the carbon retention was 0.14 wt% of the carbon source of syngas (Supplementary Table 9).

We also utilized the XPS analysis to characterize the inner surface of the Fe-SCR-*Reduced* and Fe-SCR-*Spent*. The Fe 2p peaks at 706.8 and 707.2 eV were identified on the Fe-SCR-*Reduced* and Fe-SCR-*Spent* (Supplementary Fig. 17b, c), and were attributed to iron metal and iron carbide⁴⁸⁻⁵⁰, respectively. This proved that iron metal species were transformed into iron carbide species on the inner surface of Fe-SCR during the FT synthesis. The same phenomena were also observed on the Fe-Powder-*Reduced* and Fe-Powder-*Spent* in the XPS analysis (Supplementary Fig. 17b, c). Moreover, the XRD and ⁵⁷Fe Mössbauer spectra results of Fe-Powder-*spent* have demonstrated that the γ -Fe₅C₂ was the main phase during the reaction (Supplementary Figs. 19, 20 and Supplementary Table 7). Therefore, we confirm that the grainy inner surface with the highly active γ -Fe₅C₂ on the Fe-SCR promotes the FT performance.”

Supplementary Figure 21 | SEM images and EDS linear scan analyses for the cross-section of Fe-SCR and Fe-SCR-*Spent*. (a) SEM image for the cross-section of Fe-SCR. (b) EDS linear scan analysis for the cross-section of Fe-SCR. (c) SEM image for the cross-section of Fe-SCR-*Spent*. (d) EDS linear scan analysis for the cross-section of Fe-SCR-*Spent*.

7. Figure S18 needs larger scale bars.

Response: We do appreciate the reviewer for this suggestion. We followed this suggestion, and have revised our manuscript. Please see Supplementary Figure 23 in the revised Supplementary Information.

8. The authors need to provide more information regarding the reusability experiment shown in Figure S6. Specifically, was any pretreatment performed before reusing the Fe-SCR?

Response: We thank the reviewer for this comment. In the reusability experiment, the Fe-SCR-*Spent* was calcined in air at 873 K for 60 h, and then reduced by H₂ at 673 K for 10 h with the H₂ flow rate of 40 mL min⁻¹. The pretreatments of the Fe-SCR-*Spent* were the same as those of the fresh Fe-SCR. We have added the statements in the section of "Methods", as shown below:

“Then, the catalytic systems were raised to target temperature and pressure, respectively, to start the reactions. The reusability experiment was conducted on the same reaction conditions. Before the experiment, pretreatments of the spent Fe-SCR were the same as those of the fresh Fe-SCR.”

In addition, for preparation of traditional solid catalysts, binder is often used to fabricate catalyst pellets in the industries. But the binder will lower the intrinsic catalyst regeneration and reusability. Our SCR_s simultaneously serve as reactors and catalysts, without using the binder. This is also an advantage to our SCR_s technology.

9. What is the channel volume of the different SCR designs, and why were these designs chosen? The authors only stated that they designed different SCR_s to tune the inner surface (page 5). Next the authors speculate that the design will affect the contact time of FT reactants and products. The catalyst/reactor contact time will certainly change with the channel volume if the flow rate is not accordingly adjusted. The contact time can be easily calculated given the known flow rate and the channel volume from the CAD design. How does the SCR contact time compare to the contact time in the T316/metal powder experiments? May be this

discussion can be used to generate some general design rules?

Response: We very much appreciate the reviewer for the comments and suggestions. We followed the suggestions, and calculated the inner surface and channel volume via the CAD (Rhinoceros 5.0). The results were exhibited in Supplementary Table 12 in the revised Supplementary Information. The Co-SCR-6 displayed the highest inner surface and the smallest channel volume. The different SCR designs were chosen based on the rules of easy realization, tunable geometrical structure, and high tolerance of pressure/temperature. We have added these statements in the "Results and discussion", as follows:

“.....The various morphologies of Co-SCRs were selected in the metal 3D printing, based on the rules of easy realization, tunable geometrical structure, and high tolerance of pressure/temperature. After the printing fabrications, these Co-SCRs were also applied in FT synthesis.....”

Supplementary Table 6 | Linear velocity of syngas in the T₃₁₆, Fe-Powder and Fe-SCR.^a

Sample	Inner surface (10 ⁻³ m ²)	Channel volume (10 ⁻⁶ m ³)	Linear velocity ^c (10 ⁻⁴ m s ⁻¹)
T ₃₁₆	1.2	2.2	2.8
Fe-Powder	22 ^b	/	0.2
Fe-SCR	3.9	0.6	0.8

(a) Inner surface and channel volume were obtained according to the reactor size and CAD calculation. (b) The surface of Fe-Powder was calculated based on the average particle size. (c) The linear velocity was calculated based on the equation of $V_{linear} = F_{CO+H_2} / Scat.$ (V_{linear} , F_{CO+H_2} and $Scat.$ represent linear velocity of syngas, flow rate of syngas and catalyst surface, respectively). Syngas conditions: temperature, 573 K; pressure, 1.0 MPa; flow rate, 20 ml min⁻¹.

We used linear velocity (V_{linear}) of syngas as a criterion to compare the T₃₁₆, Fe-Powder and Fe-SCR, because the traditional contact time is difficult to simultaneously describe the T₃₁₆, Fe-Powder and Fe-SCR with different spatial positions for their catalyst components. The V_{linear} was defined as translational velocity of syngas, *i.e.*, $V_{linear} = F_{CO+H_2} / Scat.$ (F_{CO+H_2} and $Scat.$ represent flow rate of syngas and catalyst surface, respectively). The V_{linear} results were displayed in Supplementary Table 6. The

Fe-Powder and the Fe-SCR exhibited lower V_{linear} than the traditional reaction tube T₃₁₆. In the revised manuscript, the statements were added in the "Results and discussion", as below:

“In addition, we also compared linear velocity (V_{linear}) of syngas for the T₃₁₆, Fe-Powder and Fe-SCR (Supplementary Table 6). The linear velocity (V_{linear}) is defined as translational velocity of syngas, i.e., $V_{linear} = F_{CO+H_2} / Scat.$ (F_{CO+H_2} and $Scat.$ represent flow rate of syngas and catalyst surface, respectively). The results unveiled that both the Fe-Powder and the Fe-SCR exhibited lower V_{linear} than the traditional reaction tube T₃₁₆.”

To compare the different Co-SCRs designs, we also calculated the linear velocity (V_{linear}) of syngas for the Co-SCRs. In addition, passage time of syngas was used as another criterion over the Co-SCRs. The passage time is defined as the time it takes for syngas to pass the channel, i.e., $T_{passage} = V_{channel} / F_{CO+H_2}$ ($V_{channel}$ and F_{CO+H_2} are the channel volume and flow rate of syngas, respectively). The results of $T_{passage}$ and V_{linear} were exhibited in Supplementary Table 12. They revealed that three key factors of inner surface, channel volume and spatial structure worked at the same time, and co-determined the liquid fuel selectivity in FT synthesis. The detailed statements and discussion have been added into the section of "Results and discussion", as follows:

“To obtain the underlying reasons of influencing the liquid fuel selectivity, we further analyzed the inner surface and channel volume, and calculated the liner velocity (V_{linear}) and passage time ($T_{passage}$) of syngas for the Co-SCRs (Supplementary Table 12). The passage time is defined as the time it takes for syngas to pass the channel, i.e., $T_{passage} = V_{channel} / F_{CO+H_2}$ ($V_{channel}$ and F_{CO+H_2} are the channel volume and flow rate of syngas, respectively). Because the F_{CO+H_2} of feed gas was constant in our FT evaluation, the high inner surface and low channel volume of Co-SCRs led to low V_{linear} and $T_{passage}$, respectively.

It is well known in FT synthesis that large catalytic surface area and low linear velocity will enhance re-adsorption of intermediate α -olefin, to promote new carbon-chain growth^{65,66}. Low channel volume and low passage time can reduce secondary reactions (hydrocracking/hydrogenolysis) of long-chain hydrocarbons^{33,67}, and keep high liquid fuel selectivity in FT process. Moreover, spatial structure of FT reactor plays a key role on regulating the balance between plug-flow and back-mixing modes of reaction gas^{68,69}. Therefore, although the Co-SCRs revealed non-linear changes on the inner surface and channel volume (Supplementary Table 12), they still

displayed a linear increase on the liquid fuel selectivity (Fig. 4b). It demonstrated that the multiple factors, as mentioned above, worked simultaneously, resulting in the non-linear, overlying phenomena of the factors. These analyses also proved that the SCR designs can provide three kinds of tunable factors, including inner surface, channel volume and spatial structure, to realize high controllability on chemical synthesis.”

Supplementary Table 12 | Inner surface, channel volume, passage time of syngas, and linear velocity of syngas for the Co-SCRs.^a

Sample	Inner surface (10 ⁻³ m ²)	Channel volume (10 ⁻⁶ m ³)	Passage time ^b (s)	Linear velocity ^c (10 ⁻⁴ m s ⁻¹)
Co-SCR-1	3.4	1.3	3.9	1.0
Co-SCR-2	3.9	1.2	3.6	0.8
Co-SCR-3	1.8	0.5	1.5	1.9
Co-SCR-4	3.1	1.1	3.3	1.1
Co-SCR	3.9	0.6	1.8	0.8
Co-SCR-5	3.2	0.8	2.4	1.0
Co-SCR-6	7.7	0.4	1.2	0.4

(a) Internal surface and channel volume was obtained by CAD calculation (Rhinoceros 5.0). (b) The passage time was calculated according to the equation of $T_{passage} = V_{channel} / F_{CO+H2}$ ($T_{passage}$, $V_{channel}$ and F_{CO+H2} represent passage time, channel volume and syngas flow rate, respectively). (c) The linear velocity was calculated based on the equation of $V_{linear} = F_{CO+H2} / Scat.$ (V_{linear} , F_{CO+H2} and $Scat.$ represent linear velocity of syngas, flow rate of syngas and catalyst surface, respectively). Syngas conditions: temperature, 533 K; pressure, 2.0 MPa; flow rate, 20 ml min⁻¹.

Reviewer #2 (Remarks to the Author):

1. The paper is describing use of Selective Laser Sintering (or Selective Laser Melting, SLM) method for preparation catalysts for petroleum refining. The basic idea here is combination of the functionality of the printed object, in this case the reaction vessel, with the functionality of the printing material, which in this case serve as catalytically active material. This is not a new idea but one of the emerging trends in 3D printing. It is possible to have considerable synergetic advances when

the object has dual functionality. It is possible to optimize the shape and size of the printed object and when it is printed with active material it is possible to select or optimize also the functionality of the material. Both of these are nicely presented in the paper.

Response: We greatly thank Reviewer 2 for the positive comments and important guidance. In the manuscript, we utilize metal 3D printing to design and manufacture self-catalytic reactors. The self-catalytic reactors, with dual functions of reactor and catalyst at high temperature and/or high pressure, can realize the conversion of C1 molecules into high value-added chemicals. We try our best to develop the multifunctional integration in metal 3D printing, and hope to stimulate the large-scale applications in various fields, such as micro-reactor assembly. We predict that this technology is able to be applied in micro-reactor area very soon, to downsize huge catalytic plants, such as FT plant, Methanol plant, without loading catalyst pellets.

2. I think this paper deserves to be published in Nature Communication. However, I think that the authors should note it the text that use of SLM-technique for printing catalytically active objects is part of a new and exciting trend in 3D printing, which aims at dual functionality of the printed object. Printing catalysts itself is not new. There are examples where FDM or similar type extruding techniques has been used to build catalytically active flow through objects: DOI: 10.1016/j.jcat.2015.11.019, DOI: 10.1016/j.cattod.2019.06.026, DOI: 10.1016/j.jclepro.2018.12.274, DOI: 10.1016/j.apcatb.2019.01.008, DOI: 10.1149/2.0341905jes, DOI: 10.1016/j.jcou.2019.07.013, DOI: 10.1021/acsapm.9b00598.

Response: We very much appreciate the reviewer for the positive comments and guidance. In the revised manuscript, we followed the guidance, and have added the statements that the 3D printing techniques of direct ink writing (DIW) and fused deposition modeling (FDM) were used in the fabrication of catalysts or reactors. The statements were also shown below:

“Recently, several research groups have also made considerable progress in catalyst preparation and reactor design^{7,8,14-30}. The 3D printing techniques, such as direct ink writing (DIW)^{15-24,28}, fused deposition modeling (FDM)^{14,16,26-28}, stereolithography (SLA)²⁹ and selective laser sintering (SLS)³⁰, were employed and developed to print the functional catalysts or reactors. The printed catalysts or reactors have exhibited many new and exciting trends for chemical synthesis and analysis.”

In addition, the relevant references (refs.) of the direct ink writing (DIW) were put in the refs. 15-24, 28; The relevant refs. of the fused deposition modeling (FDM) were put in the refs. 14, 16, 26-28. The refs. (including those mentioned by the reviewer) were also shown below:

“14. Kitson, P. J. *et al.* Digitization of multistep organic synthesis in reactionware for on-demand pharmaceuticals. *Science*, **359**, 314–319 (2018).

15. Symes M. D. *et al.* Integrated 3D-printed reactionware for chemical synthesis and analysis. *Nat. Chem.* **4**, 349–354 (2012).

16. Kitson, P. J. *et al.* 3D printing of versatile reactionware for chemical synthesis. *Nat. Protoc.* **11**, 920–936 (2016).

17. Zhu, C. *et al.* Toward digitally controlled catalyst architectures: hierarchical nanoporous gold via 3D printing. *Sci. Adv.* **4**, eaas9459 (2018).

18. Tubío, C. R. *et al.* 3D printing of a heterogeneous copper-based catalyst. *J. Catal.* **334**, 110–115 (2016).

19. Quintanilla, A. *et al.* Graphene-based nanostructures as catalysts for wet peroxide oxidation treatments: from nanopowders to 3D printed porous monoliths. *Catal. Today*, <https://doi.org/10.1016/j.cattod.2019.06.026> (2019).

20. Middelkoop, V. *et al.* Next frontiers in cleaner synthesis: 3D printed graphene-supported CeZrLa mixed-oxide nanocatalyst for CO₂ utilisation and direct propylene carbonate production. *J. Clean. Prod.* **214**, 606–614 (2019).

21. Magzoub, F. *et al.* 3D-printed ZSM-5 monoliths with metal dopants for methanol conversion in the presence and absence of carbon dioxide. *Appl. Catal. B* **245**, 486–495 (2019).

22. Middelkoop, V. *et al.* 3D printed Ni/Al₂O₃ based catalysts for CO₂ methanation - a comparative and operando XRD-CT study. *J. CO₂ Util.* **33**, 478–487 (2019).

23. Díaz-Marta, A. S. *et al.* Three-dimensional printing in catalysis: combining 3D heterogeneous copper and palladium catalysts for multicatalytic multicomponent reactions. *ACS Catal.* **8**, 392–404 (2018).

24. Azuaje, J. *et al.* An efficient and recyclable 3D printed α -Al₂O₃ catalyst for the multicomponent assembly of bioactive heterocycles. *Appl. Catal. A* **530**, 203–210 (2017).

26. Sangiorgi, A. et al. 3D printing of photocatalytic filters using a biopolymer to immobilize TiO₂ nanoparticles. *J. Electrochem. Soc.* **166**, 3239–3248 (2019).

27. Díaz-Marta, A. S. et al. Integrating reactors and catalysts through three-dimensional printing: efficiency and reusability of an impregnated palladium on silica monolith in Sonogashira and Suzuki reactions. *ChemCatChem* **12**, 1762–1771 (2020).

28. Díaz-Marta, A. S. et al. Multicatalysis combining 3D-printed devices and magnetic nanoparticles in one-pot reactions: steps forward in compartmentation and recyclability of catalysts. *ACS Appl. Mater. Interfaces* **11**, 25283–25294 (2019).”

3. In addition to these example also SLS printing has been used for building reactor parts. Similarly, idea of printing catalytic reactors or reactor parts are well known: DOI: 10.1039/C7CY00615B, DOI: 10.1021/acscatal.7b02592, DOI: 10.1021/acsomega.9b00711. The last one is an example of use of SLS-technique for printing mixer for a catalyst reactor, but even in that one it has been mentioned that it would be possible to use printing for preparing any shape of objects. This means that the catalytic activity could be added in any part of the reaction vessel, not just in the walls of the reactor as in the paper here. This approach goes far beyond the catalysis. Similar technique could be used to prepare all kind of multifunctional objects. In addition to the biological systems it has been applied in various different areas. For example, NLO active lenses, where optical properties of the lens have been combined with NLO activity of the printing material, have been printed by using SLA technique (10.1021/acsomega.8b01659).

Response: We do appreciate the reviewer for the guidance. We followed the guidance, and have added the statements that the 3D printing techniques of stereolithography (SLA) and selective laser sintering (SLS) were used in the fabrication of catalysts or reactors, as below:

“Recently, several research groups have also made considerable progress in catalyst preparation and reactor design^{7,8,14-30}. The 3D printing techniques, such as direct ink writing (DIW)^{15-24,28}, fused deposition modeling (FDM)^{14,16,26-28}, stereolithography (SLA)²⁹ and selective laser sintering (SLS)³⁰, were employed and developed to print the functional catalysts or reactors. The printed catalysts or reactors have exhibited many new and exciting trends for chemical synthesis and analysis.”

The relevant refs. of the stereolithography (SLA) technique were put in the ref. 29; The relevant refs. of the selective laser sintering (SLS) technique were put in the ref. 30. The review paper was shown in the ref. 25.

“25. Hurt, C. *et al.* Combining additive manufacturing and catalysis: a review. *Catal. Sci. Technol.* 7, 3421–3439 (2017).

29. Manzano, J. S., Wang, H. & Slowing, I. I. High throughput screening of 3d printable resins: adjusting the surface and catalytic properties of multifunctional architectures. *ACS Appl. Polym. Mater.* 1, 2890–2896 (2019).

30. Lahtinen, E. *et al.* Fabrication of porous hydrogenation catalysts by a selective laser sintering 3D printing technique. *ACS Omega* 4, 12012–12017 (2019).”

4. As a summary, I think the paper should be published in Nature communication. To my knowledge it is the first example of use of SLM printing to build catalytically active reactor vessels. It extends the use of 3D printed chemically functional objects into area of high pressure high temperature tasks. However, it should also be kept in mind that despite the obvious benefits of using metal printing it also has its limitations. It is not suitable for direct printing molecular materials i.e. molecular catalysts and it is not likely to be the first choice if the catalysis require noble metals. Therefore, the authors should put this into the wider context of latest developments in 3D-printing: multifunctional objects.

Response: We greatly thank the reviewer for the excellent guidance on 3D methodology, and do appreciate the positive comments on our manuscript. We fully agree that the authors should put the designs into the wider context of latest developments in 3D-printing. We followed the guidance, and have added the statements in the Introduction of our revised manuscript. The relevant refs. 7, 8, 14-30 were employed to support our statements. We also showed the statements, as below:

“Recently, several research groups have also made considerable progress in catalyst preparation and reactor design^{7,8,14-30}. The 3D printing techniques, such as direct ink writing (DIW)^{15-24,28}, fused deposition modeling (FDM)^{14,16,26-28}, stereolithography (SLA)²⁹ and selective laser sintering (SLS)³⁰, were employed and developed to print the functional catalysts or reactors. The printed catalysts or reactors have exhibited many new and exciting trends for chemical synthesis and analysis.”

Reviewer #3 (Remarks to the Author):

1. In my opinion, this is a very interesting article since it addresses various scientific and technological aspects currently relevant. On the one hand, 3D-printing, an emerging technology. Particularly innovative is the technique based on selective laser sintering (SLS) using metallic powder, for the elaboration of new monolithic-type reactors. On the other hand, the catalytic activity presented, very efficient and relevant. Therefore, 3D-printing fabrication of metal-based catalytic devices is a field of much interest in catalysis today.

Response: We do appreciate Reviewer 3 for the positive comments. In this work, we design and manufacture self-catalytic reactors via metal 3D printing, and utilize them in harsh reaction conditions, such as high temperature and high pressure. The reactor fabrication, coupled with catalytic function in the 3D printing, is a simple, fast method to construct catalytic system. We wish that these designs will facilitate the further development of 3D printing in the fields of chemistry and chemical engineering.

2. The degree of novelty and interest in this article is high due to several reasons: The own technique used for 3D-printing (selective laser sintering, SLS), the composition of the reactors (Fe, Ni, Co) and the type of application for which you want to apply. On the other hand, although 3D-printing technology is mostly applied to thermoplastic polymers or ceramic materials, little development still exists for metal 3D-printing, due to the high temperatures necessary in the process due to the high melting point of the metal.

Response: We greatly thank the reviewer for these positive comments. 3D printing with high degree of flexibility and freedom provides a variety of options to meet our needs. Metal 3D printing is an important type of the 3D printing technologies, but the developments are still very slow in chemical synthesis. The high melting point of metal is a weakness in the fabrication process. In our SCRs and catalysis, we try to turn the weakness into strength to realize high-pressure and high-temperature reactions.

3. Although the references provided are quite illustrative and significant within the field, I think that some more reference could be added to the text, regarding the preparation and application of monoliths containing metallic species on the surface through 3D-printing, particularly those that incorporate metal on their surface. An example is our work related to the use of monoliths in multi-catalysis (Antonio S. Díaz-Marta, ACS-Catal., 2018, 8, 392-404). The reactor-catalysts presented in this work are in fact 3D monolitos.

Response: We are very grateful to the reviewer for the suggestions. We followed the suggestions, and added the descriptions of the latest developments of 3D-printing for reactor or catalyst fabrication in the revised manuscript. The descriptions were also shown below:

“Recently, several research groups have also made considerable progress in catalyst preparation and reactor design^{7,8,14-30}. The 3D printing techniques, such as direct ink writing (DIW)^{15-24,28}, fused deposition modeling (FDM)^{14,16,26-28}, stereolithography (SLA)²⁹ and selective laser sintering (SLS)³⁰, were employed and developed to print the functional catalysts or reactors. The printed catalysts or reactors have exhibited many new and exciting trends for chemical synthesis and analysis.”

The relevant refs. were put in the refs. 7, 8, 14-30. For example, in the refs. 23, 24, 27 and 28, the direct ink writing (DIW) technique and fused deposition modeling (FDM) technique were employed to fabricate reactors or catalysts, as shown below:

“23. Díaz-Marta, A. S. *et al.* Three-dimensional printing in catalysis: combining 3D heterogeneous copper and palladium catalysts for multicomponent reactions. *ACS Catal.* **8**, 392–404 (2018).

24. Azuaje, J. *et al.* An efficient and recyclable 3D printed α -Al₂O₃ catalyst for the multicomponent assembly of bioactive heterocycles. *Appl. Catal. A* **530**, 203–210 (2017).

27. Díaz-Marta, A. S. *et al.* Integrating reactors and catalysts through three-dimensional printing: efficiency and reusability of an impregnated palladium on silica monolith in Sonogashira and Suzuki reactions. *ChemCatChem* **12**, 1762–1771 (2020).

28. Díaz-Marta, A. S. *et al.* Multicatalysis combining 3D-printed devices and magnetic nanoparticles in one-pot reactions: steps forward in compartmentation and recyclability of catalysts. *ACS Appl. Mater. Interfaces* **11**, 25283–25294 (2019).”

4. Although catalysts of this type (metal monolithics) have already been described for different catalytic transformations, in this work it is applied for industrial purposes in three different types of high value transformations: FT was performed on Fe-SCR and Co-SCR; CO₂ hydrogenation on Fe-SCR; CO₂ reforming of methane on Ni. The catalysts show a high level of reusability. That is why I consider this article as very interesting.

Response: We very much appreciate the reviewer for the positive comments. With rapid depletion of petroleum reserves, it is necessary to utilize non-petroleum resources (such as natural gas/shale gas, CO₂, biomass). Our metal 3D printing designs provide a low-cost way with high energy efficiency to improve the present industrial modes. The precise fabrication, via CAD and 3D printing, can guarantee the high reusability, and eliminate personal errors caused by different workers during catalyst preparation. Furthermore, solid catalysts often need binder to form pellets, and then loaded into industrial reactors in conventional commercial production. The binder will lower the intrinsic catalyst regeneration and reusability. But here the SCRs themselves serve as fixed and shaped catalytic sites, removing the need of binder. Therefore, elimination of personal errors and no need of binder are also two important features of our 3D SCRs technology.

5. The work seems to be very well executed. Two different centers, one in Japan and the other in China, have coordinated the manufacturing of these devices. One catalyst was prepared in Japan (Fe-SCR) and two in China (Ni-SCR and Co-SCR).

Response: We thank the reviewer very much for this comment. We have industrial and academic partners in China and Japan. Cooperation and communication are convenient in the whole project.

6. Indeed, the design integrates the concept of catalyst and reactor in the same device. The characterization of the material is very complete. However, there are some aspects of this work that have generated in me some questions or doubts:

-Since the specific surface is a key aspect in these reactors (basically it is for any

type of reactor, ceramic, metal-ceramic or ultraporous) I wonder why the authors have not provided any data about BET area (specific surface). This would help the rest of the researchers in the field to have a clear idea regarding the catalytic surface of these manufactured materials.

Response: We greatly thank the reviewer for this comment. We fully agree that the BET area is an important aspect in the reactors. In our previous work, we also performed the BET analysis, but the BET instrument cannot analyze the SCR samples, due to the low specific surface. In the revised manuscript, we improved our test method, and added SiO₂ pellet as an inner standard in the BET analysis. A mixture of SiO₂ and SCR slices was first measured, and then only the SiO₂ inner standard was measured. The BET area of SCR samples was obtained based on the difference of the mixture and SiO₂ inner standard. We have added the BET results in Supplementary Table 4.

Supplementary Table 4 | BET area for Fe-Powder and Fe-SCR before and after the pretreatments.^a

Sample	BET area (m ² g ⁻¹) ^b
Fe-Powder	12
Fe-Powder- Calcined	12
Fe-Powder- Reduced	13
Fe-Powder- Spent	13
Fe-SCR	1.6
Fe-SCR- Calcined	2.6
Fe-SCR- Reduced	2.7
Fe-SCR- Spent	3.0

(a) SiO₂ pellet was used as an inner standard for the BET analysis. (b) The SCR samples were cut into small pieces, and the weight of each piece was about 0.20~0.25 g for the BET analysis.

7. SEM images do indeed reveal a grainy surface, which is interesting from the point of view of a larger specific surface. Other data that accompany the characterization of the material such as XRD, RAMAN, Mössbauer, are very complete. In any case, I would like to know if they have considered measuring BET

area or if they have an approximate idea of the value m^2/g of catalyst.

Response: We are very grateful to the reviewer for the comments. We followed the suggestion, and added the BET analysis for the Fe-Powder and Fe-SCRs samples. As shown in Supplementary Table 4, the Fe-Powder samples, after the calcination, reduction and FT synthesis, displayed the BET area of 12~13 $m^2 g^{-1}$. The Fe-SCR samples, after the pretreatments and FT synthesis, showed the BET area of 1.6~3.0 $m^2 g^{-1}$. The Fe-SCR samples revealed lower BET areas than the Fe-Powder samples, because bulk phase of the Fe-SCRs was highly dense and did not contribute to the N_2 physisorption.

In addition, the SEM and EDS analyses, on the cross-section of Fe-SCR and Fe-SCR-*Spent*, have demonstrated that the catalytic layer was only formed on the inner surface of Fe-SCR, as shown in Supplementary Figure 21a-d. Therefore, although the Fe-SCR sample exhibited low BET area, the catalytic layer with grainy structure using in the catalytic reactions still possessed abundant porosity and large surface area (Supplementary Figure 4c, d).

8. Related to the grainy nature of the surface, it would be positive to know if the researchers carried out some type of experiment to evaluate the possible leaching of metallic particles to the reaction medium, which could eventually contaminate the liquid-fuel (ICP or similar experiment). This is not an essential question anyway, although it does give an idea of the robustness of the monolith-reactor.

Response: We followed the reviewer's suggestions, and conducted the ICP analysis for the liquid-fuels. The results were listed in Supplementary Table 10. We did not observe the metal contamination in the liquid fuels. The statement was added in the "Results and discussion" of the revised manuscript, as below:

“.....the C_{5+} selectivity could reach 65 %. The inductively coupled plasma atomic emission spectroscopy (ICP-AES) analysis demonstrated no metal contamination in the liquid fuel (Supplementary Table 10). These findings indicate that the Co-SCR not only enhances the liquid fuel selectivity, but also inhibits the formation of CO_2 by-product.....”

Supplementary Table 10 | ICP analysis for metal elements in the liquid fuel.

Sample	Metal element (mg L ⁻¹)					
	Fe	Co	Cr	Mo	W	Ni
Liquid fuel (Fe-SCR) ^a	< DL ^c	/	< DL	< DL	/	< DL
Liquid fuel (Co-SCR) ^b	/	< DL	< DL	< DL	< DL	/

(a) Liquid fuel was obtained on the Fe-SCR after FT synthesis. (b) Liquid fuel was obtained on the Co-SCR after FT synthesis. (c) Detection limit was abbreviated as DL. In the ICP analysis, we tested four metal elements for each liquid-fuel sample. They were lower than the detection limit of ICP analysis. In addition, we further conducted XRF analysis on these two samples. The results also showed that the metal elements of Fe-SCR or Co-SCR did not contaminate the liquid fuel.

9. The variability in the catalytic activity as a consequence of the design in different geometries of the reactors is discussed in the article. The selectivity of Co-SCR-6 to generate gasoline, jet fuel or diesel fuel is particularly interesting. However, from the reading at the end of the article, it is not clear to me if this variation is as a consequence of an increase in the specific surface area of the reactor or rather the geometry of the reactor itself, a longer passage time of the reagents depending on the shape, or a combination of the two factors.

Response: We do appreciate the reviewer for the positive comment and guidance. To reveal the key factors influencing the liquid fuel selectivity, we analyzed the inner surface and channel volume for the Co-SCRs, and also calculated the passage time (T_{passage}) and liner velocity (V_{linear}) of syngas on the Co-SCRs. The results were shown in Supplementary Table 12. The analyses demonstrated that the multiple factors, including inner surface, channel volume, and spatial structure, worked at the same time and co-determined the liquid fuel selectivity. In the revised manuscript, we have added the detailed statements and discussion in the "Results and discussion", as below:

“To obtain the underlying reasons of influencing the liquid fuel selectivity, we further analyzed the inner surface and channel volume, and calculated the liner velocity (V_{linear}) and passage time (T_{passage}) of syngas for the Co-SCRs (Supplementary Table 12). The passage time is defined as the time it takes for syngas to pass the channel, i.e., $T_{\text{passage}} = V_{\text{channel}} / F_{\text{CO+H}_2}$ (V_{channel} and $F_{\text{CO+H}_2}$ are

the channel volume and flow rate of syngas, respectively). Because the F_{CO+H_2} of feed gas was constant in our FT evaluation, the high inner surface and low channel volume of Co-SCRs led to low V_{linear} and $T_{passage}$, respectively.

It is well known in FT synthesis that large catalytic surface area and low linear velocity will enhance re-adsorption of intermediate α -olefin, to promote new carbon-chain growth^{65,66}. Low channel volume and low passage time can reduce secondary reactions (hydrocracking/hydrogenolysis) of long-chain hydrocarbons^{33,67}, and keep high liquid fuel selectivity in FT process. Moreover, spatial structure of FT reactor plays a key role on regulating the balance between plug-flow and back-mixing modes of reaction gas^{68,69}. Therefore, although the Co-SCRs revealed non-linear changes on the inner surface and channel volume (Supplementary Table 12), they still displayed a linear increase on the liquid fuel selectivity (Fig. 4b). It demonstrated that the multiple factors, as mentioned above, worked simultaneously, resulting in the non-linear, overlying phenomena of the factors. These analyses also proved that the SCR designs can provide three kinds of tunable factors, including inner surface, channel volume and spatial structure, to realize high controllability on chemical synthesis.”

Supplementary Table 12 | Inner surface, channel volume, passage time of syngas, and linear velocity of syngas for the Co-SCRs.^a

Sample	Inner surface (10^{-3} m^2)	Channel volume (10^{-6} m^3)	Passage time ^b (s)	Linear velocity ^c (10^{-4} m s^{-1})
Co-SCR-1	3.4	1.3	3.9	1.0
Co-SCR-2	3.9	1.2	3.6	0.8
Co-SCR-3	1.8	0.5	1.5	1.9
Co-SCR-4	3.1	1.1	3.3	1.1
Co-SCR	3.9	0.6	1.8	0.8
Co-SCR-5	3.2	0.8	2.4	1.0
Co-SCR-6	7.7	0.4	1.2	0.4

(a) Internal surface and channel volume was obtained by CAD calculation (Rhinoceros 5.0). (b) The passage time was calculated according to the equation of $T_{passage} = V_{channel} / F_{CO+H_2}$ ($T_{passage}$, $V_{channel}$ and F_{CO+H_2} represent passage time, channel volume, and flow rate of syngas, respectively). (c) The linear velocity was calculated based on the equation of $V_{linear} = F_{CO+H_2} / Scat.$ (V_{linear} , F_{CO+H_2} and $Scat.$ represent linear velocity of syngas, flow rate of syngas and catalyst surface, respectively). Syngas conditions: temperature, 533 K; pressure, 2.0 MPa; flow rate, 20 ml min⁻¹.

10. As for the virtual design (CAD) of the catalyst-reactor, the software they have used is not specified.

Response: We followed this suggestion, and added the statement in the "Method" of the revised manuscript, as follow:

“The virtual SCRs were created by computer-assisted design (CAD, Rhinoceros 5.0). The physical SCRs were prepared by metal 3D printing via a selective laser sintering.....”

11. The supplied video is very illustrative of the SLS process. I highly recommend publishing this article.

Response: We greatly thank the reviewer for the time put into the review of our manuscript, and also very much appreciate the reviewer for the precious comments and suggestions. They significantly improve our manuscript.

References

7. Parra-Cabrera, C. *et al.* 3D printing in chemical engineering and catalytic technology: structured catalysts, mixers and reactors. *Chem. Soc. Rev.* **47**, 209–230 (2018).
8. Zhou, X. & Liu, C. Three-dimensional printing for catalytic applications: current status and perspectives. *Adv. Funct. Mater.* **27**, 1701134 (2017).
14. Kitson, P. J. *et al.* Digitization of multistep organic synthesis in reactionware for on-demand pharmaceuticals. *Science*, **359**, 314–319 (2018).
15. Symes M. D. *et al.* Integrated 3D-printed reactionware for chemical synthesis and analysis. *Nat. Chem.* **4**, 349–354 (2012).
16. Kitson, P. J. *et al.* 3D printing of versatile reactionware for chemical synthesis. *Nat. Protoc.* **11**, 920–936 (2016).
17. Zhu, C. *et al.* Toward digitally controlled catalyst architectures: hierarchical nanoporous gold via 3D printing. *Sci. Adv.* **4**, eaas9459 (2018).
18. Tubío, C. R. *et al.* 3D printing of a heterogeneous copper-based catalyst. *J. Catal.* **334**, 110–115 (2016).

19. Quintanilla, A. *et al.* Graphene-based nanostructures as catalysts for wet peroxide oxidation treatments: from nanopowders to 3D printed porous monoliths. *Catal. Today*, <https://doi.org/10.1016/j.cattod.2019.06.026> (2019).
20. Middelkoop, V. *et al.* Next frontiers in cleaner synthesis: 3D printed graphene-supported CeZrLa mixed-oxide nanocatalyst for CO₂ utilisation and direct propylene carbonate production. *J. Clean. Prod.* **214**, 606–614 (2019).
21. Magzoub, F. *et al.* 3D-printed ZSM-5 monoliths with metal dopants for methanol conversion in the presence and absence of carbon dioxide. *Appl. Catal. B* **245**, 486–495 (2019).
22. Middelkoop, V. *et al.* 3D printed Ni/Al₂O₃ based catalysts for CO₂ methanation - a comparative and operando XRD-CT study. *J. CO₂ Util.* **33**, 478–487 (2019).
23. Díaz-Marta, A. S. *et al.* Three-dimensional printing in catalysis: combining 3D heterogeneous copper and palladium catalysts for multicatalytic multicomponent reactions. *ACS Catal.* **8**, 392–404 (2018).
24. Azuaje, J. *et al.* An efficient and recyclable 3D printed α -Al₂O₃ catalyst for the multicomponent assembly of bioactive heterocycles. *Appl. Catal. A* **530**, 203–210 (2017).
25. Hurt, C. *et al.* Combining additive manufacturing and catalysis: a review. *Catal. Sci. Technol.* **7**, 3421–3439 (2017).
26. Sangiorgi, A. *et al.* 3D printing of photocatalytic filters using a biopolymer to immobilize TiO₂ nanoparticles. *J. Electrochem. Soc.* **166**, 3239–3248 (2019).
27. Díaz-Marta, A. S. *et al.* Integrating reactors and catalysts through three-dimensional printing: efficiency and reusability of an impregnated palladium on silica monolith in Sonogashira and Suzuki reactions. *ChemCatChem* **12**, 1762–1771 (2020).
28. Díaz-Marta, A. S. *et al.* Multicatalysis combining 3D-printed devices and magnetic nanoparticles in one-pot reactions: steps forward in compartmentation and recyclability of catalysts. *ACS Appl. Mater. Interfaces* **11**, 25283–25294 (2019).
29. Manzano, J. S., Wang, H. & Slowing, I. I. High throughput screening of 3d printable resins: adjusting the surface and catalytic properties of multifunctional architectures. *ACS Appl. Polym. Mater.* **1**, 2890–2896 (2019).
30. Lahtinen, E. *et al.* Fabrication of porous hydrogenation catalysts by a selective laser sintering 3D printing technique. *ACS Omega* **4**, 12012–12017 (2019).

47. Jia, C.-J. *et al.* Large-scale synthesis of single-crystalline iron oxide magnetic nanorings. *J. Am. Chem. Soc.* **130**, 16968–16977 (2008).
48. Li, X. & Zhang, W. Sequestration of metal cations with zerovalent iron nanoparticles-A study with high resolution X-ray photoelectron spectroscopy (HR-XPS). *J. Phys. Chem. C* **111**, 6939-6946 (2007).
49. Butt, J. B. Carbide phases on iron-based Fischer-Tropsch synthesis catalysts part I: characterization studies. *Catal. Letters* **7**, 61–81(1990).
50. Yang, C. *et al.* Fe₃C₂ nanoparticles: a facile bromide-induced synthesis and as an active phase for Fischer–Tropsch synthesis. *J. Am. Chem. Soc.* **134**, 15814–15821 (2012).
53. Zhang, J. *et al.* Synthesis of light olefins from CO hydrogenation over Fe-Mn catalysts: effect of carburization pretreatment. *Fuel* **109**, 116–123 (2013).
54. Hamilton, N. G. *et al.* The application of inelastic neutron scattering to investigate CO hydrogenation over an iron Fischer-Tropsch synthesis catalyst. *J. Catal.* **312**, 221–231 (2014).
65. Kuipers, E. W., Vinkenburg, I. H. & Oosterbeek, H. Chain length dependence of α -olefin readsorption in Fischer-Tropsch synthesis. *J. Catal.* **155**, 137–146 (1995).
66. Iglesia, E., Reyes, S. C. & Madon, R. J. Transport-enhanced α -olefin readsorption pathways in Ru-catalyzed hydrocarbon synthesis. *J. Catal.* **129**, 238–256 (1991).
67. Sartipi, S. *et al.* Catalysis engineering of bifunctional solids for the one-step synthesis of liquid fuels from syngas: a review. *Catal. Sci. Technol.* **4**, 893–907 (2014).
68. Rai, A. *et al.* Kinetics and computational fluid dynamics study for Fischer-Tropsch synthesis in microchannel and fixed-bed reactors. *React. Chem. Eng.* **3**, 319–332 (2018).
69. Davis, B. H. Fischer-Tropsch synthesis: overview of reactor development and future potentialities. *Top. Catal.* **32**, 143–168 (2005).

REVIEWERS' COMMENTS:

Reviewer #1 (Remarks to the Author):

All reviewers' comments and suggestions have been addressed and I recommend the current version of the manuscript for publication without further modifications.

Reviewer #2 (Remarks to the Author):

I found the authors response satisfactory and think that the paper is now publishable in Nature Communications.

REVIEWERS' COMMENTS:

Reviewer #1 (Remarks to the Author):

All reviewers' comments and suggestions have been addressed and I recommend the current version of the manuscript for publication without further modifications.

Response: We greatly thank Reviewer 1 for this positive comments regarding our manuscript. The previous comments and suggestions from Reviewer 1 were very helpful to improve our manuscript. We do appreciate Reviewer 1 for the comments and suggestions.

Reviewer #2 (Remarks to the Author):

I found the authors response satisfactory and think that the paper is now publishable in Nature Communications.

Response: We very much appreciate the Reviewer 2 for the time put into the review of our manuscript. We also thank the Reviewer 2 for the previous suggestions and comments. They significantly improved our manuscript.